# Amortized Bayesian Experimental Design for Decision-Making

**Daolang Huang**
Aalto University
daolang.huang@aalto.fi

**Yujia Guo**
Aalto University
yujia.guo@aalto.fi

**Luigi Acerbi**
University of Helsinki
luigi.acerbi@helsinki.fi

**Samuel Kaski**
Aalto University
University of Manchester
samuel.kaski@aalto.fi

## Abstract

Many critical decisions, such as personalized medical diagnoses and product pricing, are made based on insights gained from designing, observing, and analyzing a series of experiments. This highlights the crucial role of experimental design, which goes beyond merely collecting information on system parameters as in traditional *Bayesian experimental design* (BED), but also plays a key part in facilitating *downstream decision-making*. Most recent BED methods use an amortized policy network to rapidly design experiments. However, the information gathered through these methods is suboptimal for down-the-line decision-making, as the experiments are not inherently designed with downstream objectives in mind. In this paper, we present an amortized decision-aware BED framework that prioritizes maximizing downstream decision utility. We introduce a novel architecture, the Transformer Neural Decision Process (TNDP), capable of instantly proposing the next experimental design, whilst inferring the downstream decision, thus effectively amortizing both tasks within a unified workflow. We demonstrate the performance of our method across several tasks, showing that it can deliver informative designs and facilitate accurate decision-making.

## 1 Introduction

In a wide array of disciplines, from clinical trials (Cheng and Shen, 2005) to medical imaging (Burger et al., 2021), a fundamental challenge is the design of experiments to infer the distribution of some unobservable, unknown properties of the systems under study. *Bayesian Experimental Design* (BED) (Lindley, 1956; Chaloner and Verdinelli, 1995; Ryan et al., 2016; Rainforth et al., 2024) is a powerful framework in this context, guiding and optimizing experimental design by maximizing the expected amount of information about parameters gained from experiments, see Fig. 1(a). However, the ultimate goal in many tasks extends beyond parameter inference to inform a *downstream decision-making* task by leveraging our understanding of these parameters. For example, in personalized medical diagnostics, a model is built based on historical data to facilitate an optimal treatment for a new patient (Bica et al., 2021). This data typically comprises patient covariates, administered treatments, and observed outcomes. Since the query of such data tends to be expensive due to, e.g., privacy issues, we need to actively design queries to optimize resource utilization. Here, when the goal is to improve decisions, the strategy of experimental designs should not focus solely on inferring the parameters of the model, but rather on guiding the final decision-making for the new patient, to ensure that each query contributes maximally to the diagnostic decision.

38th Conference on Neural Information Processing Systems (NeurIPS 2024).

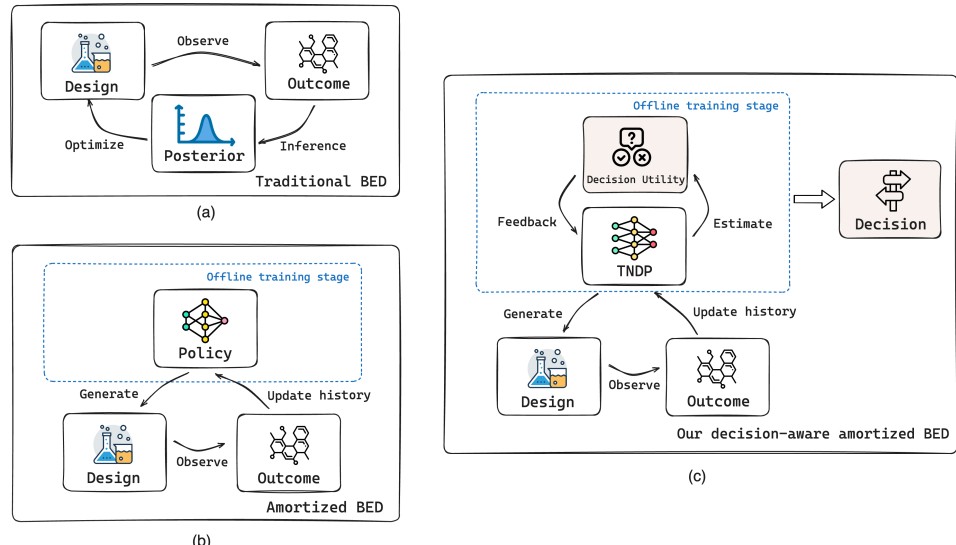

Figure 1: **Overview of BED workflows.** (a) Traditional BED, which iterates between optimizing designs, running experiments, and updating the model via Bayesian inference. (b) Amortized BED, which uses a policy network for rapid experimental design generation. (c) Our decision-aware amortized BED integrates decision utility in training to facilitate downstream decision-making.

Traditional BED methods (Rainforth et al., 2018; Foster et al., 2019, 2020; Kleinegesse and Gutmann, 2020) do not take down-the-line decision-making tasks into account during the experimental design phase. As a result, inference and decision-making processes are carried out separately, which is suboptimal for decision-making in scenarios where experiments can be adaptively designed. *Loss-calibrated inference*, which was originally introduced by Lacoste-Julien et al. (Lacoste-Julien et al., 2011) for variational approximations in Bayesian inference, provides a concept that adjusts the inference process to capture posterior regions critical for decision-making tasks. Rather than focusing on parameter estimation, the idea is to directly maximize the expected downstream utility, recognizing that accurate decision-making can proceed without exact knowledge of the posterior as not all regions of the posterior contribute equally to the downstream task. Inspired by this concept, we could consider integrating decision-making directly into the experimental design process to align the proposed designs more closely with the ultimate decision-making task.

To pick the next optimal design, standard BED methods require estimating and optimizing the expected information gain (EIG) over the design space, which can be extremely time-consuming. This limitation has led to the development of *amortized* BED (Foster et al., 2021; Ivanova et al., 2021; Blau et al., 2022, 2023), a *policy-based* method which leverages a neural network policy trained on simulated experimental trajectories to quickly generate designs, as illustrated in Fig. 1(b). Given an experiment history, these policy-based methods determine the next experimental design through a single forward pass, significantly speeding up the design process. Another significant advantage of amortized BED methods is their capacity to extract and utilize unstructured domain knowledge embedded in historical data. Unlike traditional methods, which never reuse the information from past data, amortized methods can integrate this knowledge to refine and improve design strategies for new experiments. In our problem setting, the benefits of amortization are also valuable where decisions must be made swiftly, such as when determining optimal treatment for patients in urgent settings.

In this paper, we propose an amortized decision-making-aware BED framework, see Fig. 1(c). We identify two key aspects where previous amortized BED methods fall short when applied to downstream decision-making tasks. First, the training objective of the existing methods does not consider downstream decision tasks. Therefore, we introduce the concept of *Decision Utility Gain* (DUG) to guide experimental design to better align with the downstream objective. DUG is designed to measure the improvement in the *maximum* expected utility derived from the new experiment. Second, to obtain the optimal decision, we still need to explicitly approximate the predictive distribution of the outcomes to estimate the utility. Current amortized methods learn this distribution only implicitly and therefore require extra computation for the decision-making process. To address this, we propose

a novel *Transformer neural decision process* (TNDP) architecture with dual output heads: one acting as a policy network to propose new designs, and another to approximate the predictive distribution to support the downstream decision-making. This setup allows an iterative approach where the system autonomously proposes informative experimental designs and makes optimal final decisions. Finally, since our ultimate goal is to make optimal decisions at the final stage, which may involve multiple experiments, it is crucial that our experimental designs are not myopic or overly greedy by only maximizing next-step decision utility. Therefore, we develop a non-myopic objective function that ensures decisions are made with a comprehensive consideration of future outcomes.

**Contributions.** In summary, the contributions of our work include:

- We propose the concept of *decision utility gain* (DUG) for guiding the next experimental design with a direct focus on optimizing down-the-line decision-making tasks.
- We present a novel architecture – *Transformer neural decision process* (TNDP) – designed to amortize the experimental designs directly for downstream decision-making.
- We empirically show the effectiveness of TNDP across a variety of experimental design tasks involving decision-making, where it significantly outperforms other methods that do not consider downstream decisions.

## 2 Preliminaries

### 2.1 Bayesian experimental design

BED (Lindley, 1956; Ryan et al., 2016; Rainforth et al., 2024) is a powerful statistical framework that optimizes the experimental design process. The primary goal of BED is to sequentially select a set of experimental designs $\xi \in \Xi$ and gather outcomes $y$, to maximize the amount of information obtained about the parameters of interest, denoted as $\theta$. Essentially, BED seeks to minimize the entropy of the posterior distribution of $\theta$ or, equivalently, to maximize the information that the experimental outcomes provide about $\theta$.

At the heart of BED lies the concept of *Expected Information Gain* (EIG), which quantifies the value of different experimental designs based on how much they are expected to reduce uncertainty about $\theta$, measured in terms of expected entropy ($H[\cdot]$) reduction:

$$\text{EIG}(\xi) = \mathbb{E}_{p(y|\xi)} \left[ H\left[p(\theta)\right] - H\left[p(\theta|\xi, y)\right] \right]. \tag{1}$$

The optimal design is then defined as $\xi^* = \arg\max_{\xi \in \Xi} \text{EIG}(\xi)$. In practice, calculating EIG is a computationally challenging task because it involves integrations over $p(y|\xi)$ and $p(\theta|\xi, y)$, which are both intractable. In recent years, various methods have been proposed to make the computation of the EIG feasible in practical scenarios, such as nested Monte Carlo estimators (Rainforth et al., 2018) and variational approximations (Foster et al., 2019). However, even with these advancements, the computational load remains significant, hindering feasibility in tasks that demand rapid designs. This limitation has pushed forward the development of amortized BED methods, which significantly reduce computational demands during the deployment stage.

### 2.2 Amortized BED

Amortized BED methods represent a significant shift from traditional experimental optimization techniques. Instead of optimizing for each experimental design separately, Foster et al. (2021) developed a parameterized policy $\pi$, which directly maps the experimental history $h_{1:t} = \{(\xi_1, y_1), ..., (\xi_t, y_t)\}$ to the next design $\xi_{t+1} = \pi(h_{1:t})$. To train such a policy network, Foster et al. (2021) proposed using sequential Prior Contrastive Estimation (sPCE) to optimize the lower bound of the total EIG across the entire $T$-step experiments trajectory:

$$sPCE(\pi, L) = \mathbb{E}_{p(\theta_{0:L})p(h_{1:T}|\theta_0, \pi)} \left[ \log \frac{p(h_{1:T}|\theta_0, \pi)}{\frac{1}{L+1} \sum_{\ell=0}^{L} p(h_{1:T}|\theta_\ell, \pi)} \right], \tag{2}$$

where $\theta_{1:L}$ are contrastive samples drawn from the prior distribution. Although the training of such a policy network is computationally expensive, once trained, the network can act as an oracle to quickly propose the next design through a single forward pass, thus amortizing the initial training cost over numerous deployments.

## 2.3 Bayesian decision theory

*Bayesian decision theory* (Berger, 2013) provides an axiomatic framework for decision-making under uncertainty, systematically incorporating probabilistic beliefs about unknown parameters into decision-making processes. It introduces a task-specific utility function $u(\theta, a)$, which quantifies the value of the outcomes from different decisions $a \in \mathcal{A}$ when the system is in state $\theta$. The optimal decision is then determined by maximizing the expected utility, which integrates the utility function over the unknown system parameters, given the available knowledge $h_{1:t}$:

$$a^* = \arg\max_{a \in A} \mathbb{E}_{p(\theta|h_{1:t})}[u(\theta, a)]. \tag{3}$$

In many scenarios, outcomes are stochastic and it is more typical to make decisions based on their *predictive distribution* $p(y|\xi, h_{1:t}) = \mathbb{E}_{p(\theta|h_{1:t})}[p(y|\xi, \theta, h_{1:t})]$, such as in clinical trials where the optimal treatment is chosen based on predicted patient responses rather than solely on underlying biological mechanisms. A similar setup can be found in (Kuśmierczyk et al., 2019; Vadera et al., 2021). As we switch the belief about the state of the system to the outcomes and to keep as much information as possible, we need to evaluate the effect of $\theta$ on all points of the design space. Thus, instead of the posterior over the latent state $\theta$, we represent our belief directly as $p(y_\Xi|h_{1:t}) \equiv \{p(y|\xi, h_{1:t})\}_{\xi \in \Xi}$, i.e. a joint predictive (posterior) distribution of outcomes over all possible designs given the current information $h_{1:t}$.[1] The utility is then expressed as $u(y_\Xi, a)$, which relies on the decision $a$ and all possible predicted outcomes $y_\Xi$. It is a natural extension of the traditional definition of utility by marginalizing out the posterior distribution of $\theta$. The rule of making the optimal decision is reformulated in terms of the predictive distribution as:

$$a^* = \arg\max_{a \in A} \mathbb{E}_{p(y_\Xi|h_{1:t})}[u(y_\Xi, a)]. \tag{4}$$

Traditional methods usually separate the inference and decision-making steps, which are optimal when the true posterior or the predictive distribution can be computed exactly. However, in most cases the posteriors are not directly accessible, and we often resort to using approximate distributions. This necessity results in a suboptimal decision-making process as the approximate posteriors often focus on representing the full posterior yet fail to ensure high accuracy in regions crucial for decision-making. *Loss-calibrated inference* (Lacoste-Julien et al., 2011) emerges as an effective solution to address this problem. It calibrates the inference by focusing on utility rather than mere accuracy of the approximation, thereby ensuring a more targeted posterior estimation. This method has been applied to improving Markov chain Monte Carlo (MCMC) methods (Abbasnejad et al., 2015), Bayesian neural networks (Cobb et al., 2018) and expectation propagation (Morais and Pillow, 2022).

## 3 Decision-aware BED

### 3.1 Problem setup

Our objective is to optimize the experimental design process for down-the-line decision-making. In this paper, we consider scenarios in which we design a series of experiments $\xi \in \Xi$ and observe corresponding outcomes $y$ to inform a final decision-making step. We assume we have a fixed experimental budget with $T$ query steps. For decision-making, we consider a set of possible decisions, denoted as $a \in \mathcal{A}$, with the objective of identifying an optimal decision $a^*$ that maximizes a predefined prediction-based utility function $u(y_\Xi, a)$.

### 3.2 Decision Utility Gain

Our method focuses on designing the experiments to directly improve the quality of the final decision-making. To quantify the effectiveness of each experimental design in terms of decision-making, we introduce *Decision Utility Gain* (DUG), which is defined as the difference in the expected utility of the best decision, with the new information obtained from the current experimental design, versus the best decision with the information obtained from previous experiments.

---

[1]This definition assumes conditional independence of the outcomes given the design. More generally, $p(y_\Xi|h_{1:t})$ defines a joint distribution or a *stochastic process* indexed by the set $\Xi$ (Parzen, 1999), where a familiar example could be a Gaussian process posterior defined on $\Xi \subseteq \mathbb{R}^d$ (Rasmussen and Williams, 2006).

**Definition 3.1.** Given a historical experimental trajectory $h_{1:t-1}$, the *Decision Utility Gain* (DUG) for a given design $\xi_t$ and its corresponding outcome $y_t$ at step $t$ is defined as follows:

$$\text{DUG}(\xi_t, y_t) = \max_{a \in A} \mathbb{E}_{p(y_\Xi | h_{1:t-1} \cup \{(\xi_t, y_t)\})} \left[ u(y_\Xi, a) \right] - \max_{a \in A} \mathbb{E}_{p(y_\Xi | h_{1:t-1})} \left[ u(y_\Xi, a) \right]. \quad (5)$$

DUG measures the improvement in the *maximum* expected utility from observing a new experimental design, differing in this from standard marginal utility gain (see e.g., Garnett, 2023). The optimal design is the one that provides the largest increase in maximal expected utility. Shifting from parameter-centric to utility-centric evaluation, we directly evaluate the design's influence on the decision utility, bypassing the need to reduce the uncertainty of unknown latent parameters.

At the time we choose the design $\xi_t$, the outcome remains uncertain. Therefore, we should consider the *Expected Decision Utility Gain* (EDUG) with respect to the marginal distribution of the outcomes to select the next design.

**Definition 3.2.** The *Expected Decision Utility Gain* (EDUG) for a design $\xi_t$, given the historical experimental trajectory $h_{1:t-1}$, is defined as:

$$\text{EDUG}(\xi_t) = \mathbb{E}_{p(y_t | \xi_t, h_{1:t-1})} \left[ \text{DUG}(\xi_t, y_t) \right]. \quad (6)$$

With EDUG, we can guide the experimental design without calculating the posterior distribution. If we knew the true predictive distribution, we could always determine the one-step lookahead optimal design by maximizing EDUG across the design space with $\xi^* = \arg\max_{\xi \in \Xi} \text{EDUG}(\xi)$. However, in practice, the true predictive distributions are often unknown, making the optimization of EDUG exceptionally challenging. This difficulty arises due to the inherent bi-level optimization problem and the need to evaluate two layers of expectations, both over the unknown predictive distribution.

To avoid the expensive computational cost of optimizing EDUG, we propose leveraging a policy network, inspired by Foster et al. (2021), that directly maps historical data to the next design. This approach sidesteps the continuous need to optimize EDUG by learning a design strategy over many simulated experiment trajectories during the training phase. It can dramatically reduce computational demands at deployment, allowing for more efficient real-time decisions.

## 4 Amortizing decision-aware BED

A fully amortized BED framework for decision-making requires not only amortizing the experimental design but also the predictive distribution to approximate the expected utility. Moreover, *permutation invariance* is often assumed in sequential BED (Foster et al., 2021)[2], meaning that the sequence of experiments does not influence the cumulative information gained. Conditional neural processes (CNPs) (Garnelo et al., 2018; Nguyen and Grover, 2022; Huang et al., 2023b) provide a suitable basis for developing our framework due to their design, which not only respects the permutation invariance of the inputs by treating them as an unordered set but also amortizes modeling of the predictive distributions. See Appendix A for a brief introduction to CNPs and TNPs.

### 4.1 Transformer Neural Decision Processes

The architecture of our model, termed *Transformer Neural Decision Process* (TNDP), is a novel architecture building upon the Transformer neural process (TNP) (Nguyen and Grover, 2022). It aims to amortize both the experimental design and the predictive distributions for the subsequent decision-making process. The data architecture of our system comprises four parts $D = \{D^{(c)}, D^{(p)}, D^{(q)}, \text{GI}\}$:

- A **context set** $D^{(c)} = h_{1:t} = \{(\xi_i^{(c)}, y_i^{(c)})\}_{i=1}^{t}$ contains all past $t$-step designs and outcomes.

- A **prediction set** $D^{(p)} = \{(\xi_i^{(p)}, y_i^{(p)})\}_{i=1}^{n_p}$ consists of $n_p$ design-outcome pairs used for approximating $p(y_\Xi | h_{1:t})$, which is closely related to the training objective of the CNPs. The output from this head can then be used to estimate the expected utility.

- A **query set** $D^{(q)} = \{\xi_i^{(q)}\}_{i=1}^{n_q}$ consists of $n_q$ candidate experimental designs being considered for the next step. In scenarios where the design space $\Xi$ is continuous, we randomly sample a set of query points for each iteration during training. In the deployment phase, optimal experimental designs can be obtained by optimizing the model's output.

---

[2]When permutation invariance does not hold in some cases, our model can be easily adapted by adding positional encoding to the input.

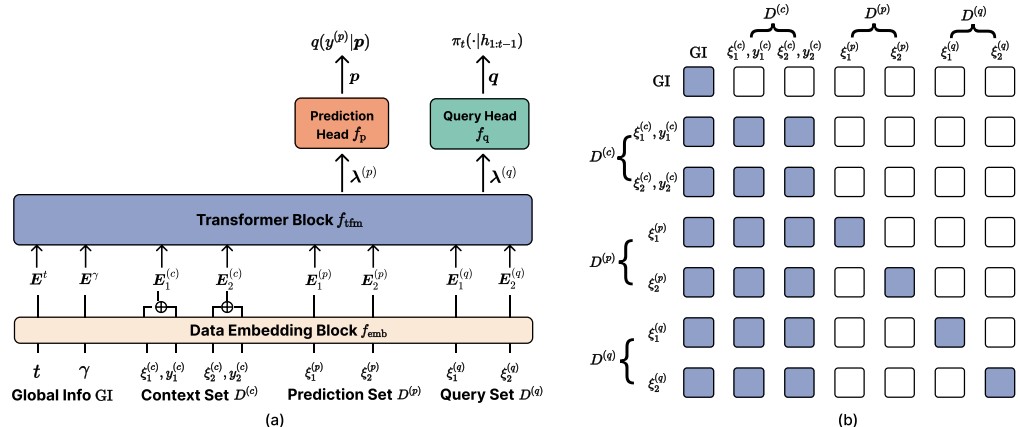

Figure 2: **Illustration of TNDP.** (a) An overview of TNDP architecture with input consisting of 2 observed design-outcome pairs from $D^{(c)}$, 2 designs from $D^{(p)}$ for prediction, and 2 candidate designs from $D^{(q)}$ for query. (b) The corresponding attention mask. The colored squares indicate that the elements on the left can attend to the elements on the top in the self-attention layer of $f_{\text{tfm}}$.

- **Global information** GI $= [t, \gamma]$ where $t$ represents the current step in the experimental sequence, and $\gamma$ encapsulates task-related information, which could include contextual data relevant to the decision-making process. We will further explain the choice of $\gamma$ in Section 6.

TNDP comprises four main components: the data embedder block $f_{\text{emb}}$, the Transformer block $f_{\text{tfm}}$, the prediction head $f_{\text{p}}$, and the query head $f_{\text{q}}$. Each component plays a distinct role in the overall decision-aware BED process. The full architecture is shown in Fig. 2(a).

At first, each set of $D$ is processed by the **data embedder block** $f_{\text{emb}}$ to map to an aligned embedding space. These embeddings are then concatenated to form a unified representation $\boldsymbol{E} = \text{concat}(\boldsymbol{E}^{(c)}, \boldsymbol{E}^{(p)}, \boldsymbol{E}^{(q)}, \boldsymbol{E}^{\text{GI}})$. Please refer to Appendix B for a detailed explanation of how we embed the data. After the initial embedding, the **Transformer block** $f_{\text{tfm}}$ processes $\boldsymbol{E}$ using self-attention mechanisms to produce a single attention matrix, which is subsequently processed by an attention mask (see Fig. 2(b)) that allows for selective interactions between different data components, ensuring that each part contributes appropriately to the final output. To explain, each design from the prediction set $D^{(p)}$ is configured to attend to itself, the global information, and the historical data, reflecting the dependence of the predictions on the historical data and the independence from other designs. Similarly, each $\xi^{(q)}$ in the query set $D^{(q)}$ is also restricted to attend only to itself, the global information, and the historical data. This setup preserves the independence of each candidate design, ensuring that the evaluation of one design neither influences nor is influenced by others. The output of $f_{\text{tfm}}$ is then split according to the specific needs of the query and prediction head $\boldsymbol{\lambda} = [\boldsymbol{\lambda}^{(p)}, \boldsymbol{\lambda}^{(q)}] = f_{\text{tfm}}(\boldsymbol{E})$.

The primary role of the **prediction head** $f_{\text{p}}$ is to approximate $p(y_\Xi | h_{1:t})$ with a family of parameterized distributions $q(y_\Xi | \boldsymbol{p}_t)$, where $\boldsymbol{p}_t = f_{\text{p}}(\boldsymbol{\lambda}_t^{(p)})$ is the output of $f_{\text{p}}$ at the step $t$. The training of $f_{\text{p}}$ is by minimizing the negative log-likelihood of the predicted probabilities:

$$\mathcal{L}^{(p)} = -\sum_{t=1}^{T}\sum_{i=1}^{n_{\text{p}}} \log q(y_i^{(p)} | \boldsymbol{p}_{i,t}) = -\sum_{t=1}^{T}\sum_{i=1}^{n_{\text{p}}} \log \mathcal{N}(y_i^{(p)} | \boldsymbol{\mu}_{i,t}, \boldsymbol{\sigma}_{i,t}^2), \tag{7}$$

where $\boldsymbol{p}_{i,t}$ represents the output of design $\xi_i^{(p)}$ at step $t$. Here we choose a Gaussian likelihood with $\boldsymbol{\mu}$ and $\boldsymbol{\sigma}$ representing the predicted mean and standard deviation split from $\boldsymbol{p}$.

The **query head** $f_{\text{q}}$ processes the transformed embeddings $\boldsymbol{\lambda}^{(q)}$ from the Transformer block to generate a policy distribution over possible experimental designs. Specifically, it converts the embeddings into a probability distribution used to select the next experimental design. The outputs of the query head, $\boldsymbol{q} = f_{\text{q}}(\boldsymbol{\lambda}^{(q)})$, are mapped to a probability distribution via a Softmax function:

$$\pi(\xi_{j,t}^{(q)} | h_{1:t-1}) = \frac{\exp(\boldsymbol{q}_{j,t})}{\sum_{i=0}^{n_{\text{q}}} \exp(\boldsymbol{q}_{i,t})}, \tag{8}$$

where $\boldsymbol{q}_{j,t}$ represents the $t$-step's output for the candidate design $\xi_j^{(q)}$.

To design a reward signal that guides $f_q$ in proposing informative designs, we first define a single-step immediate reward based on DUG (Eq. (5)), replacing the true predictive distribution with our approximated distribution:

$$r_t(\xi_t^{(q)}) = \max_{a \in A} \mathbb{E}_{q(y_\Xi | \boldsymbol{p}_t)} \left[ u(y_\Xi, a) \right] - \max_{a \in A} \mathbb{E}_{q(y_\Xi | \boldsymbol{p}_{t-1})} \left[ u(y_\Xi, a) \right]. \tag{9}$$

This reward quantifies how the experimental design influences our decision-making by estimating the improvement in expected utility that results from incorporating new experimental outcomes. However, this objective remains myopic, as it does not account for the future or the final decision-making. To address this, we employ the REINFORCE algorithm (Williams, 1992), which allows us to consider the impact of the current design on future rewards. The final loss of $f_q$ can be written as the negative expected reward for the complete experimental trajectory:

$$\mathcal{L}^{(q)} = -\sum_{t=1}^{T} R_t \log \pi(\xi_t^{(q)} | h_{1:t-1}), \tag{10}$$

where $R_t = \sum_{k=t}^{T} \alpha^{k-t} r_k(\xi_k^{(q)})$ represents the non-myopic discounted reward. The discount factor $\alpha$ is used to decrease the importance of rewards received at later time step. $\xi_t^{(q)}$ is obtained through sampling from the policy distribution $\xi_t^{(q)} \sim \pi(\cdot | h_{1:t-1})$.

The update of $f_q$ depends critically on the accuracy with which $f_p$ approximates the predictive distribution. Ultimately, the effectiveness of decision-making relies on the informativeness of the designs proposed by $f_q$, ensuring that every step in the experimental trajectory is optimally aligned with the overarching goal of maximizing the decision utility. The full algorithm of our method is shown in Appendix C.

## 5 Related work

Lindley (1972) proposes the first decision-theoretic BED framework, later reiterated by Chaloner and Verdinelli (1995). However, their utility is defined based on individual designs, while our utility is formulated in terms of a stochastic process and is designed for the final decision-making task after multiple rounds of experimental design. Recently, several other BED frameworks that focus on different downstream properties have been proposed. Bayesian Algorithm Execution (BAX) (Neiswanger et al., 2021) develops a BED framework that optimizes experiments based on downstream properties of interest. BAX introduces a new metric that queries the next experiment by maximizing the mutual information between the property and the outcome. CO-BED (Ivanova et al., 2023) introduces a contextual optimization method within the BED framework, where the design phase incorporates information-theoretic objectives specifically targeted at optimizing contextual rewards. Neiswanger et al. (2022) presents an information-based acquisition function for Bayesian optimization which explicitly considers the downstream task. Zhong et al. (2024) proposes a goal-oriented BED framework for nonlinear models using MCMC to optimize the EIG on predictive quantities of interest. Filstroff et al. (2024) presents a framework for active learning that queries data to reduce the uncertainty on the posterior distribution of the optimal downstream decision.

In recent years, various amortized BED methods have emerged. Foster et al. (2021) is the first to introduce this framework; subsequent work extends it to scenarios with unknown likelihood (Ivanova et al., 2021) and improved performance using reinforcement learning (Blau et al., 2022; Lim et al., 2022). The latest research proposes a semi-amortized framework that periodically updates the policy during the experiment to improve adaptability (Ivanova et al., 2024). Maraval et al. (2024) proposes a fully amortized Bayesian optimization (BO) framework that employs a similar TNP architecture, while their work focuses specifically on BO objectives, our approach addresses general downstream decision-making tasks. Additionally, our framework introduces a novel coupled training objective between query and prediction heads, providing a more integrated architecture for decision-making.

Our proposed architecture is based on pre-trained Transformer models. Transformer-based neural processes (Müller et al., 2021; Nguyen and Grover, 2022; Chang et al., 2024) serve as the foundational structure for our approach, but they have not considered experimental design. Decision Transformers (Chen et al., 2021; Zheng et al., 2022) can be used for sequentially designing experiments. However, we additionally amortize the predictive distribution, making the learning process more challenging.

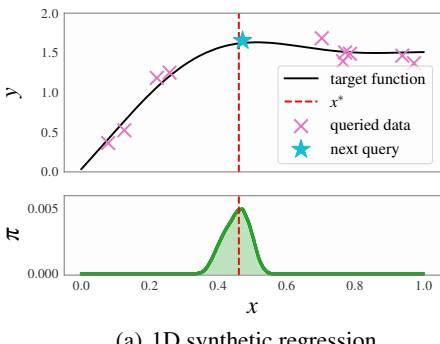

(a) 1D synthetic regression

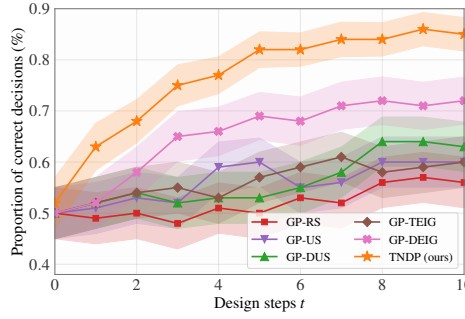

(b) Decision-aware active learning

Figure 3: **Results of synthetic regression and decision-aware active learning.** (a) The top figure represents the true function and the initial known points. The red line indicates the location of $x^*$. The blue star marks the next query point, sampled from the policy's predicted distribution shown in the bottom figure. (b) Mean and standard error of the proportion of correct decisions on 100 test points w.r.t. the acquisition steps. Our TNDP significantly outperforms other methods.

## 6    Experiments

In this section, we evaluate our proposed framework on several tasks. Our experimental approach is detailed in Appendix B. In Appendix F.3, we provide additional ablation studies of TNDP to show the effectiveness of our query head and the non-myopic objective function. The code to reproduce our experiments is available at `https://github.com/huangdaolang/amortized-decision-aware-bed`.

### 6.1    Toy example: synthetic regression

We begin with an illustrative example to show how our TNDP works. We consider a 1D synthetic regression task where the goal is to perform regression at a specific test point $x^*$ on an unknown function. To accurately predict this point, we need to sequentially design some new points to query. This example can be viewed as a prediction-oriented active learning (AL) task (Smith et al., 2023).

The design space $\Xi = \mathcal{X}$ is the domain of $x$, and $y$ is the corresponding noisy observations of the function. Let $\mathcal{Q}(\mathcal{X})$ denote the set of combinations of distributions that can be output by TNDP, we can then define decision space to be $\mathcal{A} = \mathcal{Q}(\mathcal{X})$. The downstream decision is to output a predictive distribution for $y^*$ given a test point $x^*$, and the utility function $u(y_\Xi, a) = \log q(y^*|x^*, h_{1:t})$ is the log probability of $y$ under the predicted distribution, given the queried historical data $h_t$.

During training, we sample functions from Gaussian Processes (GPs) (Rasmussen and Williams, 2006) with squared exponential kernels of varying output variances and lengthscales and randomly sample a point as the test point $x^*$. We set the global contextual information $\lambda$ as the test point $x^*$. For illustration purposes, we consider only the case where $T = 1$. Additional details for the data generation can be found in Appendix E.

**Results.** From Fig. 3(a), we can observe that the values of $\pi$ concentrate near $x^*$, meaning our query head $f_q$ tends to query points close to $x^*$ to maximize the DUG. This is an intuitive example demonstrating that our TNDP can adjust its design strategy based on the downstream task.

### 6.2    Decision-aware active learning

We now show another application of our method in a case of decision-aware AL studied by Filstroff et al. (2024). In this experiment, the model will be used for downstream decision-making after performing AL, i.e., we will use the learned information to take an action towards a specific target. A practical application of this problem is personalized medical diagnosis introduced in Section 1, where a doctor needs to query historical patient data to decide on a treatment for a new patient.

We use the same problem setup as in Filstroff et al. (2024). The decision space consists of $N_d$ available decisions, $a \in \mathcal{A} \in \{1, ..., N_d\}$. The design space $\Xi = \mathcal{X} \times \mathcal{A}$ is composed of the patient's

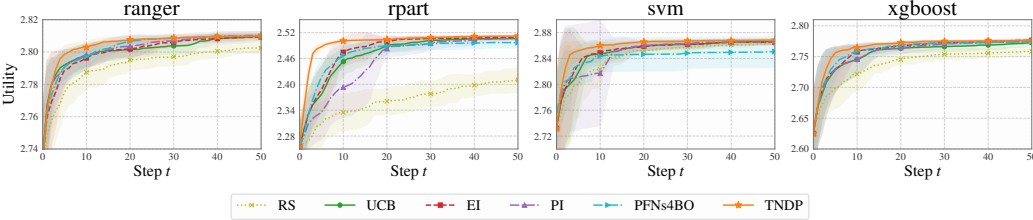

Figure 4: **Results on Top-$k$ HPO task.** For each meta-dataset, we calculated the average utility across all available test sets. The error bars represent the standard deviation over five runs. TNDP consistently achieved the best performance in terms of utility.

covariates $x$ and the decisions $a$ they receive. The outcome $y$ is the treatment effect after the patient receives the decision, which is influenced by real-world unknown parameters $\theta$ such as medical conditions. For historical data, each patient is associated with only one decision. The utility function $u(y_\Xi, a) = \mathbb{I}(\hat{a}^*, a^*)$ is a binary accuracy score that measures whether we can make the correct decision for a new patient $x^*$ based on the queried history, where $\hat{a}^*$ is the predicted Bayesian optimal decision and $a^*$ the true optimal decision. Here, $u(y_\Xi, a) = 1$ if and only if $\hat{a}^* = a^*$.

In our experiment, we use the synthetic dataset from Filstroff et al. (2024), the details of the data generating process can be found in Appendix F. We set $N_d = 4$ and use independent GPs to generate different outcomes. Each data point is randomly assigned a decision, and the outcome is the corresponding $y$ value from the associated GP. We randomly select a test point $x^*$ and determine the optimal decision $a^*$ based on the GP that provides the maximum $y$ value at $x^*$. We set global contextual information $\lambda$ as the covariates of the test point $x^*$.

We compare TNDP with other non-amortized AL methods: random sampling (GP-RS), uncertainty sampling (GP-US), decision uncertainty sampling (GP-DUS), targeted information (GP-TEIG) introduced by Sundin et al. (2018), and decision EIG (GP-DEIG) proposed by Filstroff et al. (2024). A detailed description of each method can be found in Appendix F. Each method is tested on 100 different $x^*$ points, and the average utility, i.e., the proportion of correct decisions, is calculated.

**Results.** The results are shown in Fig. 3(b), where we can see that TNDP achieves significantly better average accuracy than other methods. Additionally, we conduct an ablation study of TNDP in Appendix F.3 to verify the effectiveness of $f_q$. We further analyze the deployment running time to show the advantage of amortization, see Appendix D.1.

## 6.3 Top-$k$ hyperparameter optimization

In traditional optimization tasks, we typically only aim to find a single point that maximizes the underlying function $f$. However, instead of identifying a single optimal point, there are scenarios where we wish to estimate a set of top-$k$ distinct optima. For example, this is the case in robust optimization, where selecting multiple points can safeguard against variations in data or model performance.

In this experiment, we choose hyperparameter optimization (HPO) as our task and conduct experiments on the HPO-B datasets (Arango et al., 2021). The design space $\Xi \subseteq \mathcal{X}$ is a finite set defined over the hyperparameter space and the outcome $y$ is the accuracy of a given configuration on a specific dataset. Our decision is to find $k$ hyperparameter sets, denoted as $a = (a_1, ..., a_k) \in A \subseteq \mathcal{X}^k$, with $a_i \neq a_j$. The utility function is then defined as $u(y_\Xi, a) = \sum_{i=1}^{k} y_{a_i}$, where $y_{a_i}$ is the accuracy corresponding to the hyperparameter configuration $a_i$. In this experiment, the global contextual information $\lambda = \emptyset$.

We compare our methods with five different BO methods: random sampling (RS), Upper Confidence Bound (UCB), Expected Improvement (EI), Probability of Improvement (PI), and an amortized method PFNs4BO (Müller et al., 2023), which is a transformer-based model designed for hyperparameter optimization. We set $k = 3$ and $T = 50$, starting with an initial dataset of 5 points. Our experiments are conducted on four search spaces selected from the HPO-B benchmark. All results

are evaluated on a predefined test set, ensuring that TNDP does not encounter these test sets during training. For more details, see Appendix G.

**Results.** From the experimental results (Fig. 4), our method demonstrates superior performance across all four meta-datasets, particularly during the first 10 queries, achieving clearly better utility gains. This indicates that our TNDP can effectively identify high-performing hyperparameter configurations early in the optimization process.

Finally, we included a real-world experiment on retrosynthesis planning. Specifically, our task is to assist chemists in identifying the top-$k$ synthetic routes for a novel molecule, as selecting the most practical routes from many random routes generated by the retrosynthesis software can be troublesome. The detailed description of the task and the results are shown in Appendix G.3.

## 7 Discussion

### 7.1 Limitations & future work

We recognize that the training of the query head inherently poses a reinforcement learning (RL) (Li, 2017) problem. Currently, we employ a basic REINFORCE algorithm, which can result in unstable training, particularly for tasks with sparse reward signals. For more complex problems in the future, we could deploy advanced RL methods, such as Proximal Policy Optimization (PPO) (Schulman et al., 2017); the trade-offs include the introduction of additional hyperparameters and increased computational cost. Like all amortized approaches, our method requires a large amount of data and upfront training time to develop a reliable model. Besides, our architecture is based on the Transformer, which suffers from quadratic complexity with respect to the input sequence length. This can become a bottleneck when the query set is very large. Future work could focus on designing more sample-efficient methods to reduce the data and training requirements. Our TNDP follows the common practice in the neural processes literature (Garnelo et al., 2018) of using independent Gaussian likelihoods. If modeling correlations between points is crucial for the downstream task, we can replace the output with a joint multivariate normal distribution (Markou et al., 2022) or predict the output autoregressively (Bruinsma et al., 2023). Following most BED approaches, our work assumes that the model is well-specified. However, model misspecification or shifts in the utility function during deployment could impact the performance of the amortized model (Rainforth et al., 2024). Future work could address the challenge of robust experimental design under model misspecification (Huang et al., 2023a). Another limitation is that our system is currently constrained to accepting designs of the same dimensionality. Future work could focus on developing dimension-agnostic methods to expand the scope of amortization. Lastly, our model is trained on a fixed-step length, assuming a finite horizon for the experimental design process. Future research could explore the design of systems that can handle infinite horizon cases, potentially improving the applicability of TNDP to a broader range of real-world problems.

### 7.2 Conclusions

In this paper, we proposed an amortized framework for decision-aware Bayesian experimental design (BED). We introduced the concept of *Decision Utility Gain* (DUG) to guide experimental design more effectively toward optimizing decision outcomes. Towards amortization, we developed a novel *Transformer Neural Decision Process* (TNDP) architecture with dual output heads: one for proposing new experimental designs and another for approximating the predictive distribution to facilitate optimal decision-making. Our experimental results demonstrated that TNDP significantly outperforms traditional BED methods across a variety of tasks. By integrating decision-making considerations directly into the experimental design process, TNDP not only accelerates the design of experiments but also improves the quality of the decisions derived from these experiments.

**Acknowledgements**

DH, LA and SK were supported by the Research Council of Finland (Flagship programme: Finnish Center for Artificial Intelligence FCAI). YG was supported by Academy of Finland grant 345604. LA was also supported by Research Council of Finland grants 358980 and 356498. SK was also supported by the UKRI Turing AI World-Leading Researcher Fellowship, [EP/W002973/1]. The authors wish to thank Aalto Science-IT project, and CSC–IT Center for Science, Finland, for the computational and data storage resources provided.

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

# Appendix

The appendix is organized as follows:

## A  Conditional neural processes

CNPs (Garnelo et al., 2018) are designed to model complex stochastic processes through a flexible architecture that utilizes a *context set* and a *target set*. The context set consists of observed data points that the model uses to form its understanding, while the target set includes the points to be predicted. The traditional CNP architecture includes an encoder and a decoder. The encoder is a DeepSet architecture to ensure permutation invariance, it transforms each context point individually and then aggregates these transformations into a single representation that captures the overall context. The decoder then uses this representation to generate predictions for the target set, typically employing a Gaussian likelihood for approximation of the true predictive distributions. Due to the analytically tractable likelihood, CNPs can be efficiently trained through maximum likelihood estimation.

### A.1  Transformer neural processes

Transformer Neural Processes (TNPs), introduced by Nguyen and Grover (2022), improve the flexibility and expressiveness of CNPs by incorporating the Transformer's attention mechanism (Vaswani et al., 2017). In TNPs, the transformer architecture uses self-attention to process the context set, dynamically weighting the importance of each point. This allows the model to create a rich representation of the context, which is then used by the decoder to generate predictions for the target set. The attention mechanism in TNPs facilitates the handling of large and variable-sized context sets, improving the model's performance on tasks with complex input-output relationships. The Transformer architecture is also useful in our setups where certain designs may have a more significant impact on the decision-making process than others. For more details about TNPs, please refer to Nguyen and Grover (2022).

## B  Implementation details

### B.1  Embedders

The embedder $f_{\text{emb}}$ is responsible for mapping the raw data to a space of the same dimension. For the toy example and the top-$k$ hyperparameter task, we use three embedders: a design embedder $f_{\text{emb}}^{(\xi)}$, an outcome embedder $f_{\text{emb}}^{(y)}$, and a time step embedder $f_{\text{emb}}^{(t)}$. Both $f_{\text{emb}}^{(\xi)}$ and $f_{\text{emb}}^{(y)}$ are multi-layer perceptions (MLPs) with the following architecture:

- **Hidden dimension**: the dimension of the hidden layers, set to 32.
- **Output dimension**: the dimension of the output space, set to 32.
- **Depth**: the number of layers in the neural network, set to 4.
- **Activation function**: ReLU is used as the activation function for the hidden layers.

The time step embedder $f_{\text{emb}}^{(t)}$ is a discrete embedding layer that maps time steps to a continuous embedding space of dimension 32.

For the decision-aware active learning task, since the design space contains both the covariates and the decision, we use four embedders: a covariate embedder $f_{\text{emb}}^{(x)}$, a decision embedder $f_{\text{emb}}^{(d)}$, an outcome embedder $f_{\text{emb}}^{(y)}$, and a time step embedder $f_{\text{emb}}^{(t)}$. $f_{\text{emb}}^{(x)}$, $f_{\text{emb}}^{(y)}$ and $f_{\text{emb}}^{(t)}$ are MLPs which use the same settings as described above. The decision embedder $f_{\text{emb}}^{(d)}$ is another discrete embedding layer.

For context embedding $\boldsymbol{E}^{(\text{c})}$, we first map each $\xi_i^{(\text{c})}$ and $y_i^{(\text{c})}$ to the same dimension using their respective embedders, and then sum them to obtain the final embedding. For prediction embedding $\boldsymbol{E}^{(\text{p})}$ and query embedding $\boldsymbol{E}^{(\text{q})}$, we only encode the designs. For $\boldsymbol{E}^{\text{GI}}$, except the embeddings of the time step, we also encode the global contextual information $\lambda$ using $f_{\text{emb}}^{(x)}$ in the toy example and the decision-aware active learning task. All the embeddings are then concatenated together to form our final embedding $\boldsymbol{E}$.

## B.2 Transformer blocks

We utilize the official `TransformerEncoder` layer of PyTorch (Paszke et al., 2019) (https://pytorch.org) for our transformer architecture. For all experiments, we use the same configuration: the model has 6 Transformer layers, with 8 heads per layer, the MLP block has a hidden dimension of 128, and the embedding dimension size is set to 32.

## B.3 Output heads

The prediction head, $f_{\text{p}}$ is an MLP that maps the Transformer's output embeddings of the query set to the predicted outcomes. It consists of an input layer with 32 hidden units, a ReLU activation function, and an output layer. The output layer predicts the mean and variance of a Gaussian likelihood, similar to CNPs.

For the query head $f_{\text{q}}$, all candidate experimental designs are first mapped to embeddings $\boldsymbol{\lambda}^{(\text{q})}$ by the Transformer, and these embeddings are then passed through $f_{\text{q}}$ to obtain individual outputs. We then apply a Softmax function to these outputs to ensure a proper probability distribution. $f_{\text{q}}$ is an MLP consisting of an input layer with 32 hidden units, a ReLU activation function, and an output layer.

## B.4 Training details

For all experiments, we use the same configuration to train our model. We set the initial learning rate to `5e-4` and employ the cosine annealing learning rate scheduler. The number of training epochs is set to 50,000 for top-$k$ tasks and 100,000 for other tasks, and the batch size is 16. For the REINFORCE, we use a discount factor of $\alpha = 0.99$.

## C   Full algorithm for training TNDP

---
**Algorithm 1** Transformer Neural Decision Processes (TNDP)

---
1: **Input:** Utility function $u(y_\Xi, a)$, prior $p(\theta)$, likelihood $p(y|\theta, \xi)$, query horizon $T$
2: **Output:** Trained TNDP
3: **while** within the training budget **do**
4:  $\quad$ Sample $\theta \sim p(\theta)$ and initialize $D$
5:  $\quad$ **for** $t = 1$ to $T$ **do**
6:  $\quad\quad$ $\xi_t^{(\text{q})} \sim \pi_t(\cdot|h_{1:t-1})$ $\qquad\qquad\qquad\qquad\qquad\qquad$ ▷ sample next design from policy
7:  $\quad\quad$ Sample $y_t \sim p(y|\theta, \xi)$ $\qquad\qquad\qquad\qquad\qquad\qquad\quad$ ▷ observe outcome
8:  $\quad\quad$ Set $h_{1:t} = h_{1:t-1} \cup \{(\xi_t^{(\text{q})}, y_t)\}$ $\qquad\qquad\qquad\qquad$ ▷ update history
9:  $\quad\quad$ Set $D^{(\text{c})} = h_{1:t}$, $D^{(\text{q})} = D^{(\text{q})} \setminus \{\xi_t^{(\text{q})}\}$ $\qquad\qquad\quad$ ▷ update $D$
10: $\quad\quad$ Calculate $r_t(\xi_t^{(\text{q})})$ with $u(y_\Xi, a)$ using Eq. (9) $\qquad\quad$ ▷ calculate reward
11: $\quad$ **end for**
12: $\quad$ $R_t = \sum_{k=t}^{T} \alpha^{k-t} r_k(\xi_k^{(\text{q})})$ $\qquad\qquad\qquad\qquad\qquad$ ▷ calculate cumulative reward
13: $\quad$ Update TNDP using $\mathcal{L}^{(\text{p})}$ (Eq. (7)) and $\mathcal{L}^{(\text{q})}$ (Eq. (10))
14: **end while**
15: At deployment, we can use $f^{(\text{q})}$ to sequentially query $T$ designs. Afterward, based on the queried experiments, we perform one-step final decision-making using the prediction from $f^{(\text{p})}$.

---

# D Computational cost analysis

## D.1 Inference time analysis

We evaluate the inference time of our algorithm during the deployment stage. We select decision-aware active learning as the experiment for our time comparison. All experiments are evaluated on an `Intel Core i7-12700K` CPU. We measure both the acquisition time and the total time. The acquisition time refers to the time required to compute one next design, while the total time refers to the time required to complete 10 rounds of design. The final results are presented in Table A1, with the mean and standard deviation calculated over 10 runs.

Traditional methods rely on updating the GP and optimizing the acquisition function, which is computationally expensive. D-EIG and T-EIG require many model retraining steps to get the next design, which is not tolerable in applications requiring fast decision-making. However, since our model is fully amortized, once it is trained, it only requires a single forward pass to design the experiments, resulting in significantly faster inference times.

| Method | Acquisition time (s) | Total time (s) |
|---|---|---|
| GP-RS | 0.00002(0.00001) | 28(7) |
| GP-US | 0.07(0.01) | 29(7) |
| GP-DUS | 0.38(0.02) | 44(5) |
| T-EIG (Sundin et al., 2018) | 1558(376) | 15613(3627) |
| D-EIG (Filstroff et al., 2024) | 572(105) | 5746(1002) |
| TDNP (ours) | 0.015(0.004) | 0.31(0.06) |

Table A1: Comparison of computational costs across different methods. We report the mean value and (standard deviation) derived from 10 runs with different seeds.

## D.2 Overall training time

Throughout this paper, we carried out all experiments, including baseline model computations and preliminary experiments not included in the final paper, on a GPU cluster featuring a combination of Tesla P100, Tesla V100, and Tesla A100 GPUs. We estimate the total computational usage to be roughly 5000 GPU hours. For each experiment, it takes around 10 GPU hours on a Tesla V100 GPU with 32GB memory to reproduce the result, with an average memory consumption of 8 GB.

# E Details of toy example

## E.1 Data generation

In our toy example, we generate data using a GP with the Squared Exponential (SE) kernel, which is defined as:

$$k(x, x') = v \exp \left( -\frac{(x - x')^2}{2\ell^2} \right),$$

(A1)

where $v$ is the variance, and $\ell$ is the lengthscale. Specifically, in each training iteration, we draw a random lengthscale $\ell \sim 0.25 + 0.75 \times U(0, 1)$ and the variance $v \sim 0.1 + U(0, 1)$, where $U(0, 1)$ denotes a uniform random variable between 0 and 1.

# F Details of decision-aware active learning experiments

## F.1 Data generation

For this experiment, we use a GP with a Squared Exponential (SE) kernel to generate our data. The covariates $x$ are drawn from a standard normal distribution. For each decision, we use an independent GP to simulate different outcomes. In each training iteration, the lengthscale for each GP is randomly sampled as $\ell \sim 0.25 + 0.75 \times U(0, 1)$ and the variance as $v \sim 0.1 + U(0, 1)$, where $U(0, 1)$ denotes a uniform random variable between 0 and 1.

## F.2 Other methods description

We compare our method with other non-amortized approaches, all of which use GPs as the functional prior. Each model is equipped with an SE kernel with automatic relevance determination. GP hyperparameters are estimated with maximum marginal likelihood.

Our method is compared with the following methods:

- Random sampling (GP-RS): randomly choose the next design $\xi_t$ from the query set.

- Uncertainty sampling (GP-US): choose the next design $\xi_t$ for which the predictive distribution $p(y_t|\xi_t, h_{t-1})$ has the largest variance.

- Decision uncertainty sampling (GP-DUS): choose the next design $\xi_t$ such that the predictive distribution of the optimal decision corresponding to this design is the most uncertain.

- Targeted information (GP-TEIG) (Sundin et al., 2018): a targeted active learning criterion, introduced by (Sundin et al., 2018), selects the next design $\xi_t$ that provides the highest EIG on $p(y^*|x^*, h_{t-1} \cup \{(\xi_t, y_t)\})$.

- Decision EIG (GP-DEIG) (Filstroff et al., 2024): choose the next design $\xi_t$ which directly aims at reducing the uncertainty on the posterior distribution of the optimal decision. See Filstroff et al. (2024) for a detailed explanation.

## F.3 Ablation study

We also carry out an ablation study to verify the effectiveness of our query head and the non-myopic objective function. We first compare TNDP with TNDP using random sampling (TNDP-RS), and the results are shown in Fig. A1(a). We observe that the designs proposed by the query head significantly improve accuracy, demonstrating that the query head can propose informative designs based on downstream decisions.

We also evaluate the impact of the non-myopic objective by comparing TNDP with a myopic version that only optimizes for immediate utility rather than long-term gains ($\alpha = 0$). The results, presented in Fig. A1(b), show that TNDP with the non-myopic objective function achieves higher accuracy across iterations compared to using the myopic objective. This indicates that our non-myopic objective effectively captures the long-term benefits of each design choice, leading to improved overall performance.

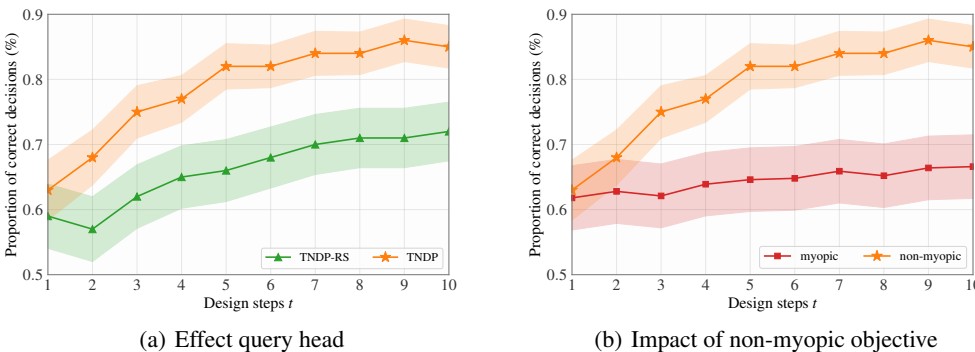

(a) Effect query head

(b) Impact of non-myopic objective

Figure A1: **Comparison of TNDP variants on the decision-aware active learning task.** (a) Shows the effect of the query head, where TNDP outperforms TNDP-RS, demonstrating its ability to generate informative designs. (b) Illustrates the impact of the non-myopic objective, with TNDP achieving higher accuracy than the myopic version.

# G  Details of top-$k$ hyperparameter optimization experiments

## G.1  Data

In this task, we use HPO-B benchmark datasets (Arango et al., 2021). The HPO-B dataset is a large-scale benchmark for HPO tasks, derived from the OpenML repository. It consists of 176 search spaces (algorithms) evaluated on 196 datasets, with a total of 6.4 million hyperparameter evaluations. This dataset is designed to facilitate reproducible and fair comparisons of HPO methods by providing explicit experimental protocols, splits, and evaluation measures.

We extract four meta-datasets from the HPOB dataset: ranger (id=7609, $d_x$=9), svm (id=5891, $d_x$=8), rpart (id=5859, $d_x$=6), and xgboost (id=5971, $d_x$=16). In the test stage, the initial context set is chosen based on their pre-defined indices. For detailed information on the datasets, please refer to https://github.com/releaunifreiburg/HPO-B.

## G.2  Other methods description

In our experiments, we compare our method with several common acquisition functions used in HPO. We use GPs as surrogate models for these acquisition functions. All the implementations are based on BoTorch (Balandat et al., 2020) (https://botorch.org/). The acquisition functions compared are as follows:

- **Random Sampling (RS)**: This method selects hyperparameters randomly from the search space, without using any surrogate model or acquisition function.
- **Upper Confidence Bound (UCB)**: This acquisition function balances exploration and exploitation by selecting points that maximize the upper confidence bound. The UCB is defined as:
$$\alpha_{\text{UCB}}(\mathbf{x}) = \mu(\mathbf{x}) + \kappa\sigma(\mathbf{x}), \tag{A2}$$
where $\mu(\mathbf{x})$ is the predicted mean, $\sigma(\mathbf{x})$ is the predicted standard deviation, and $\kappa$ is a parameter that controls the trade-off between exploration and exploitation.
- **Expected Improvement (EI)**: This acquisition function selects points that are expected to yield the greatest improvement over the current best observation. The EI is defined as:
$$\alpha_{\text{EI}}(\mathbf{x}) = \mathbb{E}[\max(0, f(\mathbf{x}) - f(\mathbf{x}^+))], \tag{A3}$$
where $f(\mathbf{x}^+)$ is the current best value observed, and the expectation is taken over the predictive distribution of $f(\mathbf{x})$.
- **Probability of Improvement (PI)**: This acquisition function selects points that have the highest probability of improving over the current best observation. The PI is defined as:
$$\alpha_{\text{PI}}(\mathbf{x}) = \Phi\left(\frac{\mu(\mathbf{x}) - f(\mathbf{x}^+) - \omega}{\sigma(\mathbf{x})}\right), \tag{A4}$$
where $\Phi$ is the cumulative distribution function of the standard normal distribution, $f(\mathbf{x}^+)$ is the current best value observed, and $\omega$ is a parameter that encourages exploration.

In addition to those non-amortized GP-based methods, we also compare our method with an amortized surrogate model **PFNs4BO** (Müller et al., 2023). It is a Transformer-based model designed for hyperparameter optimization which does not consider the downstream task. We use the pre-trained PFNs4BO-BNN model which is trained on HPO-B datasets and choose PI as the acquisition function, the model and the training details can be found in their official implementation (https://github.com/automl/PFNs4BO).

## G.3  Additional experiment on retrosynthesis planning

We now show a real-world experiment on retrosynthesis planning (Blacker et al., 2011). Specifically, our task is to help chemists identify the top-$k$ synthetic routes for a novel molecule (Mo et al., 2021), as it can be challenging to select the most practical routes from many random options generated by the retrosynthesis software (Stevens, 2011; Szymkuć et al., 2016). In this task, the design space for each molecule $m$ is a finite set of routes that can synthesize the molecule. The sequential experimental

design is to select a route for a specific molecule to query its score $y$, which is calculated based on the tree edit distance (Bille, 2005) from the best route. The downstream task is to recommend the top-$k$ routes with the highest target-specific scores based on the collected information.

In this experiment, we choose $k = 3$ and $T = 10$, and set the global information $\gamma = m$. We train our TNDP on a novel non-public meta-dataset, including 1500 molecules with 70 synthetic routes for each molecule. The representation dimension of the molecule is 64 and that of the route is 264, both of which are learned through a neural network. Given the high-dimensional nature of the data representation, it is challenging to directly compare TNDP with other GP-based methods, as GPs typically struggle with scalability and performance in such high-dimensional settings. Therefore, we only compare TNDP with TNDP using random sampling. The final results are evaluated on 50 test molecules that are not seen during training, as shown in Fig. A2.

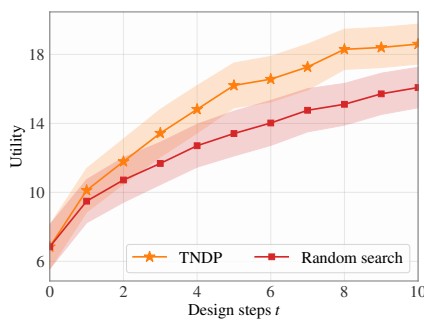

Figure A2: **Results of retrosynthesis planning experiment.** The utility is the sum of the quality scores of top-$k$ routes and is calculated with 50 molecules. Our TNDP outperforms the random search baseline.

