# OpenReview forum: "Amortized Bayesian Experimental Design for Decision-Making"
_NeurIPS.cc/2024/Conference — NeurIPS 2024 poster_

### Official Review · Reviewer_7z19 · 2024-07-01

**Soundness:** 3
**Presentation:** 2
**Contribution:** 4
**Rating:** 7
**Confidence:** 4

**Summary:**

This paper proposes a method for decision-aware Bayesian experimental design, where the design is not optimized with respect to the most accurate posterior distribution of the latent parameters but rather with respect to the expected utility gain of the actual (down-stream) decision task.

**Strengths:**

This is an innovative paper with high practical relevance. The proposed method appears sound and the corresponding neural networks well designed to suit the goal. Despite my questions and concerns (see below), I am positive about this paper overall and eager to increase my score should my points be addressed.

**Weaknesses:**

- The presentation of p(y_Xi | h_t) between Eq 3 and 4 is partially unclear to me. From the definition, it seems this is not actually a distribution but a set of distributions. To me, then notation p(y_Xi | h_t) appears to be quite the abuse of notation because we cannot readily read this it as a single distribution. Can you perhaps think about a different notation that makes this easier to parse and understand? Relatedly, in Equation 4, it appears that we compute an expectation over p(y_Xi | h_t). But how do we compute an expectation over a set of distributions? I think I get what the authors do and want to imply but to me this notation doesn’t help in understanding it.
- Equation 7: It seems we approximate the predictive distribution always by a Gaussian. I mean this of course works if the true underlying function is some kind of GP, but what if the true predictive distribution is far away from Gaussian? I don’t see this choice to be discussed properly so I consider it a weakness of this paper for now.
- The discussion of training and inference time can only be found in the appendix. Specifically, training speed seems to be substantial, which of course makes sense for an amortized method. However, I don’t see any discussion for when the training actually amortizes. That is, how many BED tasks do we need to run at minimum before the total (training + “inference”) time of the new method becomes better than those of the competing methods. More generally, I think a discussion of speed should be more prominent in the paper.
- 6.1 toy example was hard for me to understand at first. Is this just a standard BO task to find the point where the unknown function is maximal?

**Questions:**

- In 4.1 Query set: How problematic is the fact that we randomly generate some designs from the design space. Doesn’t this mean we need a distribution over the design space? How can we obtain (or define) such a distribution in general?
- In 4.1 Query set: You say that in the deployment phase we can obtain the optimal design by optimizing the models (which model’s?) output. How do you optimize this exactly?
- Given that (non-decision aware) amortized BED methods exist, why are the benchmarks only comparing against non-amortized methods? I suggest to also add amortized methods to the benchmarks unless you can convince me that this is not sensible for some reason.
- What is the scalability of the method in terms of all relevant dimensions, e.g., dimensionality of xi, y, a, etc?
- Figure 4: you say that your method provides substantial gains, but at least on the scale in the figure, gains seem to be small. Can you clarify why you feel that the improvements are indeed “substantial gains”?
- The method has quite a lot of components, I wonder which of the components is responsible for the improved results? For example, how relevant is it to consider non-myopic designs, i.e., how does the method perform when only trained in a myopic setup? Relatedly, are the alternative methods myopic or non-myopic?

**Limitations:**

The paper discusses several limitations. I am missing a discussion on the initial overhead of training, which is usually substantial in amortized methods.

---

> ### Author Rebuttal · Authors · 2024-08-07
>
> Thank you for your positive comments and thoughtful questions. We address your remarks and questions below.
>
> > 1. The presentation of p(y_Xi | h_t) between Eq 3 and 4 is partially unclear to me…
>
> Thanks for the question. $p(y_\Xi | h_t)$ is a joint distribution and is well-defined as a stochastic process. Please refer to the global response for further clarification.
>
> > 2. What if the true predictive distribution is far away from Gaussian?
>
> That’s a good point. Please refer to the response for question 9 of reviewer WHLG.
>
> > 3. The discussion of training and inference time can only be found in the appendix…I think a discussion of speed should be more prominent in the paper.
>
> The key advantage of amortized methods lies in their efficiency during the inference phase. While the upfront training time can be substantial, this investment pays off when the model is used repeatedly for multiple BED tasks.
> To answer your specific question about when the training amortizes. For example, in our active learning experiment, we observed that traditional methods like DEIG require approximately 100 minutes for a complete BED task. In contrast, our model's total training time is around 10 hours, with the inference time being negligible. Therefore, after conducting more than 6 experiments during the deployment phase, the cumulative cost of our method becomes more efficient than the traditional method.
>
> > 4. Toy example was hard for me to understand at first. Is this just a standard BO task?
>
> Thanks for the question. The toy experiment is not a standard BO task but a special active learning task. Our goal is to estimate the value of an unknown function at a specific target point $x^*$ by actively querying points. Traditional active learning methods are not target-oriented and typically query points based on the overall uncertainty of the function, which is not optimal for estimating the value at $x^*$. Our method, however, considers the downstream task, meaning our ultimate goal is to perform regression at the target location $x^*$. Thus, our decision-making-aware policy will strategically query points, specifically around $x^*$, as shown in Figure 3(a).
>
> > 5. How problematic is the fact that we randomly generate some designs from the design space?
>
> That’s a good question. For most problems, we usually either have some prior knowledge about the design space or the candidate designs come from a predefined finite set, as is commonly practiced in the pool-based active learning literature [3]. The more prior knowledge we have about the problem, the better the candidate designs we can generate.
>
> > 6. You say that in the deployment phase we can obtain the optimal design by optimizing the models (which model’s?) output. How do you optimize this exactly?
>
> We apologize for the unclear description. For continuous design spaces, we can optimize $\xi^{(q)}$ to maximize the query head's output $\mathbf{q}$, thereby obtaining the design with the highest probability. This can be achieved using optimization techniques such as gradient ascent.
>
> > 7. Given that (non-decision aware) amortized BED methods exist, why are the benchmarks only comparing against non-amortized methods?
>
> We have included a new amortized method baseline in the top-$k$ experiments and also explained why we didn’t compare with any amortized BED method previously. Please refer to the response for question 5 of reviewer WHLG.
>
> > 8. What is the scalability of the method in terms of all relevant dimensions?
>
> We have validated our method's effectiveness on high-dimensional design spaces, e.g., in the top-$k$ optimization task, $d_x$ for ranger is 9, and 16 for xgboost. Regarding the output dims, our paper focuses on single-output cases, the same as most BED works. However, our architecture can be readily generalized to multidimensional outputs by adding separate prediction heads for different outputs, as demonstrated in CNP literature [4, 5]. Lastly, for the action dims, we believe this does not significantly affect the performance since the action is made after the model's predictions.
>
> > 9. Figure 4: Can you clarify why you feel that the improvements are indeed “substantial gains”?
>
> We acknowledge that the term "substantial" may have been an overstatement, we will revise the wording. However, it is clear that our method outperforms the other baselines, particularly within the first 10 queries.
>
> > 10. I wonder which of the components is responsible for the improved results? How does the method perform when only trained in a myopic setup? Relatedly, are the alternative methods myopic or non-myopic?
>
> The improved results of our method are primarily due to our query head. This module amortizes the experimental design process, which learns common features from different tasks, allowing it to propose more valuable experiments. We conducted an ablation study in Appendix F.3 to verify the performance improvement brought by the query head.
>
> Regarding the difference in performance between non-myopic and myopic objectives, we conducted an additional ablation study, as detailed in our global response. All the alternative methods we considered in our paper are myopic.
>
> > 11. I am missing a discussion on the initial overhead of training.
>
> Thanks for your suggestion. Indeed, like all amortized approaches, our method requires a large amount of data and upfront training time to develop a reliable model. However, once the model is trained, it offers long-term benefits of faster inference. We will discuss this trade-off in the revised manuscript.
>
> **References**
>
> [1] Garnelo et al. (2018). Conditional neural processes. *ICML*.
>
> [2] Müller et al. (2021). Transformers can do Bayesian inference. *ICLR*.
>
> [3] Settles (2012). Active learning. *Springer*.
>
> [4] Markou et al. (2022). Practical conditional neural processes via tractable dependent predictions. *ICLR*.
>
> [5] Bruinsma et al. (2023). Autoregressive conditional neural processes. *ICLR*.

---

> > ### Comment · Reviewer_7z19 · 2024-08-09
> >
> > I thank the authors for their thoughtful responses and additional experiments, I have rasied my score from 6 to 7.

---

> > > ### Author Response · Authors · 2024-08-12
> > >
> > > Thank you for your response and for increasing your score. We really appreciate it.

---

### Official Review · Reviewer_Ctfm · 2024-07-07

**Soundness:** 2
**Presentation:** 3
**Contribution:** 3
**Rating:** 6
**Confidence:** 5

**Summary:**

The paper looks at the problem of designing Bayesian optimal experiments taking into account the downstream decision making. At the core is a Transformer Neural Decision Process (TNDP) architecture that is trained to amortise the experimental design process whilst simultaneously inferring the optimal downstream decision.

**Strengths:**

- Relevant and interesting topic: Downstream decision making is what ultimately matters, so taking this into account when designing experiments to collect data can result in more cost- and sample-efficient learning.

- Motivation for the paper as well as clarity of writing are excellent. Contextualisation relative to prior work can be improved as outlined in the next section.

- The proposed Transformer Neural Decision Process (TNDP) architecture is tailored to the BED problem, is well-explained and adds some novelty to the architectures typically used in the field.

**Weaknesses:**

### Sections 2.2 & 3.2 and Lindley's decision-theoretic BED [1]:

My main issue with the paper is the presentation of DUG and EDUG as novel. This framework was first formulated in [1], and is very well summarised in Section 1.3 of [2]. I strongly recommend the authors read that section, and present their Section 3.2 accordingly, acknowledging they follow Lindley, 1972. The questions/comments in the next 2 bullets are a consequence of this omission of literature.

- Second paragraph of Sec 2.2: I am not sure how the predictive distribution $p(y | \xi, h_t)$ is defined. I would think it is $p(y | \xi, h_t) = \mathbb{E}_{p(\theta |h_t)} [p(y | \xi, \theta)]$. Whether or not you compute/approximate the posterior $p(\theta |h_t)$, or seek to directly approximate $p(y | \xi, h_t)$ (eg variationally), I think you should explicitly define what this quantity is.

- I am not sure how the utility $u(y_\Xi, a)$ is defined. From a Bayesian decision-theoretic approach, the utility has to depend on the state of the world $\theta$, as well as the experiments $\xi$ you are going to perform (which I guess is implicit in $y_\Xi$). So shouldn't the "lowest level" utility be a function $u(y, \theta, \xi, a)$, which you then integrate over $p(\theta|h_t)$, to obtain $u(y, \xi, a) = \mathbb{E}_{p(\theta|h_t)} [u(y, \theta, \xi, a)]$, then take $\max$ wrt $a$,  and finally integrate over the predictive $p(y |\xi, h_t)$ to obtain an expected utility, which can then act a design ranking criterion, as you do in Eq 4 and (cf Eq 2 in [2]).

### Related work:

For a field that has such rich history and renewed interest from the ML community recently, the related works section is quite short and sparse on citations. Some areas that are missing include:
- Decision-theoretic BED: as previously discussed, the general framework of utility-based BED was developed by Lindley (1972).
- BED + RL: this work touches on some aspects of RL; It might be good to discuss relations recent works in the intersection such as [5] and [6] (in addition to those mentioned)
- Decision-theoretic approaches in related fields such as Bayesian Optimisation, e.g. [7], [8]
- Finally, I'm not too familiar with this line of literature, but  more recent work around decision transformers---is there any relation between TNDP with works like [9] and [10]?

### Other:

- Line 6: "most recent BED methods use amortised inference with a policy network" is not quite correct in the sense that no "real inference" (posterior updates on the parameters $\theta$) are performed.
- Line 179: "to ensure the framework satisfied the permutation invariance property of sequential BED": not all BED problems are permutation invariant. For example, designing experiments for time series models (e.g SIR in [3] and [4]), permutation invariance does not hold. This aspect has been discussed in e.g. Section 3.3 of [3].
- Assuming you do want a permutation invariant architecture (most design problems fall in that category): by conditioning on $t$ as part of the global information (GI) set, I think you actually break that invariance. This is because encoding $(\xi, y)$ at time $t$ or at time $s$ will give you different outputs. As far as I can tell from Fig2b), $D_c$ does attend to GI. Could you please explain if that's the case or I have misunderstood something?

-----
#### References

[1] Lindley, D. V. (1972). Bayesian statistics: A review. Society for industrial and applied mathematics.

[2] Chaloner, K., & Verdinelli, I. (1995). Bayesian experimental design: A review. Statistical science, 273-304.

[3] Ivanova, D. R., Foster, A., Kleinegesse, S., Gutmann, M. U., & Rainforth, T. (2021). Implicit deep adaptive design: Policy-based experimental design without likelihoods. Advances in neural information processing systems, 34, 25785-25798.

[4] Kleinegesse, S., & Gutmann, M. U. (2019, April). Efficient Bayesian experimental design for implicit models. In The 22nd International Conference on Artificial Intelligence and Statistics (pp. 476-485). PMLR.

[5] Mehta, V., Paria, B., Schneider, J., Ermon, S., & Neiswanger, W. (2021). An experimental design perspective on model-based reinforcement learning. arXiv preprint arXiv:2112.05244.

[6] Mehta, V., Char, I., Abbate, J., Conlin, R., Boyer, M., Ermon, S., ... & Neiswanger, W. (2022). Exploration via planning for information about the optimal trajectory. Advances in Neural Information Processing Systems, 35, 28761-28775.

[7] Neiswanger, W., Yu, L., Zhao, S., Meng, C., & Ermon, S. (2022). Generalizing Bayesian optimization with decision-theoretic entropies. Advances in Neural Information Processing Systems, 35, 21016-21029.

[8] Ivanova, D. R., Jennings, J., Rainforth, T., Zhang, C., & Foster, A. (2023, July). CO-BED: information-theoretic contextual optimization via Bayesian experimental design. In International Conference on Machine Learning (pp. 14445-14464). PMLR.

[9] Chen, L., Lu, K., Rajeswaran, A., Lee, K., Grover, A., Laskin, M., ... & Mordatch, I. (2021). Decision transformer: Reinforcement learning via sequence modeling. Advances in neural information processing systems, 34, 15084-15097.

[10] Zheng, Q., Zhang, A., & Grover, A. (2022, June). Online decision transformer. In international conference on machine learning (pp. 27042-27059). PMLR.

**Questions:**

In addition to the questions raised in the Weaknesses section:

1.  I think the main contribution of the paper is the TNDP architecture. Have the authors performed any ablations, e.g. not sharing the same embedding block? Not including $t$ in the GI?
2. In the decision-aware AL experiment: why does the random baseline perform as good as all the other ones?
3. Could you give guidance on choosing utility functions? For the experiments in the paper it is quite straightforward to define them, but in real-world practical application that might not be the case. This is the reason why the mutual information has become the de facto standard utility in BED.

**Limitations:**

Some limitations of the work were outlined in the Discussion section of the paper. Regarding negative societal impact, the field of experimental design (which boils down to efficient data collection), generally warrants some discussion.

The experiments presented in this paper mostly use synthetic data and do not have negative impact; the HPO experiment, which uses real data does not (directly) represent an application with negative impact. However, applying these methods in real-world applications, particularly if decisions directly affect humans, as in e.g. personalised medicine, could raise concerns around bias, fairness, explainability and privacy.

I would suggest to the authors to add 1-2 sentences in their limitations section to acknowledge 1) the synthetic or semi-synthetic nature of the experiments, and 2) potential concerns that might arise when applying their method in real-world applications.

---

> ### Author Rebuttal · Authors · 2024-08-07
>
> Thank you for your detailed review and the valuable references you provided. We address your questions and points raised below.
>
> **Weaknesses**
>
> > 1. My main issue with the paper is the presentation of DUG and EDUG as novel.
>
> We greatly appreciate your provided references and insights. We will include a discussion in Sections 2.2 and 3.2 about the connections and distinctions between our work and the decision-theoretic BED frameworks mentioned in [1, 2]. For further details, please refer to the global response and our answers to the following two questions.
>
> > 2. I am not sure how the predictive distribution $p(y|\xi, h_t)$ is defined… I think you should explicitly define what this quantity is.
>
> Yes, $p(y|\xi, h_t) = \mathbb{E}_{p(\theta|h_t)}[p(y|\xi, \theta, h_t)]$. It is obtained by marginalizing the posterior distribution of the parameters. In our architecture, we directly approximate the predictive distribution with neural processes, thereby bypassing the need to define and approximate the underlying parameters. We will add a description of the predictive distribution in P4L126 to clarify this point.
>
> > 3. I am not sure how the utility $u(y_{\Xi}, a)$ is defined.
>
> Our utility function is defined in terms of outcomes, which slightly differs from the traditional definition. The distribution of outcomes is obtained by (implicitly) marginalizing over $\theta$. A similar decision-theoretical setup can be found in [3]. As mentioned in the global response, the major difference between EDUG and Equation 2 in [2] is that we include an additional expectation over $p(y_\Xi | h_t)$, as the optimal Bayes action in our setup is based on the predictions of outcomes.
>
> > 4. The related works section is quite short and sparse on citations.
>
> Thanks for your references, we will add them. Regarding the differences with decision transformers, while both our architecture and theirs are based on transformers, their "decision" is similar to the step of experimental design in our context. However, we additionally amortize the downstream decision-making process, making the learning process more challenging.
>
> > 5. Line 6: "most recent BED methods use amortised inference with a policy network" is not quite correct in the sense that no "real inference" are performed.
>
> Agreed. We will change to “most recent BED methods leverage an amortized policy network to rapidly design experiments”.
>
> > 6. Line 179: "to ensure the framework satisfied the permutation invariance property of sequential BED": not all BED problems are permutation invariant.
>
> Agreed, we acknowledge that permutation invariance was meant more as an assumption than an inherent property of BED. We will fix it and add that for tasks where permutation invariance does not hold, our model can be easily adapted by incorporating positional encoding to add sequential/temporal information to the design.
>
> > 7. Assuming you do want a permutation invariant architecture… I think you actually break that invariance… Could you please explain if that's the case or I have misunderstood something?
>
> That is a misunderstanding. Permutation invariance here only means that changing the order of the historical data does not affect the design of the next experiment. Our architecture satisfies that. (Note that even without the timestamp information, the encoding as such would give different outputs at different points of time.)
>
> **Questions**
>
> > 8. Have the authors performed any ablations?
>
> We conducted an experiment in Appendix F.3 to verify the effectiveness of the query head. Additionally, we have added a new set of ablation experiments on the discount factor; please refer to the global response for details. Regarding the inclusion of time $t$, we performed ablation studies across different dimensions. The results were inconsistent. Since we only had time to run our algorithm once for each dimension, we cannot draw definitive conclusions. We will further investigate the role of $t$.
>
> > 9. In the decision-aware AL experiment: why does the random baseline perform as good as all the other ones?
>
> Because methods like US and DUS are traditional active learning methods which do not take the decision-making problem into account. Hence there is no guarantee that the queries made would improve the quality of the decision-making. However, we can see that DEIG actually performs better than the random baseline, as it considers the decision-making problem by taking into account the posterior distribution of the optimal decision for the target point.
>
> > 10. Could you give guidance on choosing utility functions?
>
> Our utility function comes directly from the downstream decision-making task. We suspect the popularity of mutual information arises from an unwillingness to commit to a specific task, which makes sense if the task is not known. However, if we know our downstream task well, such as in drug design where we aim to maximize the efficacy of the drug while minimizing its risks, we can leverage input from domain experts to define our objectives. We will comment on this in the Discussion.
>
> **Limitations**
>
> > 11. I would suggest to the authors to add 1-2 sentences to acknowledge 1) the synthetic or semi-synthetic nature of the experiments, and 2) potential concerns that might arise when applying their method in real-world applications.
>
> Thanks. We conducted a new retrosynthesis planning experiment with real-world data; please refer to the global response. We will also discuss the potential negative societal impact, especially when experiments or decisions directly affect humans, in the limitations section.
>
> **References**
>
> [1] Lindley, D. V. (1972). Bayesian statistics: A review. *Society for industrial and applied mathematics*.
>
> [2] Chaloner et al. (1995). Bayesian experimental design: A review. *Statistical science*.
>
> [3] Kuśmierczyk et al. (2019). Variational Bayesian decision-making for continuous utilities. *Neurips*.

---

> > ### Comment · Reviewer_Ctfm · 2024-08-09
> >
> > I acknowledge I have seen the rebuttal and will respond in detail. Unfortunately, this will likely happen over the weekend.

---

> > > ### Comment · Reviewer_Ctfm · 2024-08-12
> > >
> > > I thank the authors for their rebuttal and for clarifying the majority of my questions. I've increased my score by a point.

---

> > > > ### Author Response · Authors · 2024-08-12
> > > >
> > > > Thank you for your reply and for raising the score. We greatly appreciate it.

---

### Official Review · Reviewer_WHLG · 2024-07-09

**Soundness:** 3
**Presentation:** 3
**Contribution:** 3
**Rating:** 6
**Confidence:** 4

**Summary:**

The paper proposes a transformer-based architecture for jointly sampling designs and decisions in Bayesian Experiment Design (BED) using a forward-looking criterion. The latter considers the improvement in maximum expected utility brought about by a new design-outcome pair, where the expectation is taken with respect to the predictive distribution of the model. The main innovation of the paper lies in the coupling between information gain and utility maximization in an amortized, transformer-based framework in the spirit of attentive neural processes. The performance of the new architecture is evaluated on a toy regression task and two more representative models, exhibiting stable performance gains over contender methods.

**Strengths:**

- The paper is clearly written, the ideas and formulations are stringent and well-justified, overall making it easy to follow and a pleasure to read (with the exception of Section 4.1, see below).

- The proposed architecture and training objectives are novel and seem to unlock both qualitative and quantitative improvements over existing methods.

- The results indicate superior and stable performance of the proposed architecture on two interesting tasks, along a toy 1D GP model which seems to be a standard proof-of-concept task in the neural process (NP) literature.

**Weaknesses:**

- Some notational confusion can be avoided by consistently using the notation $a_{1:t}$ to denote a sequence of $t$ elements and $a_t$ to denote the $t$-th element in the sequence. Currently, $h_t$ denotes a sequence, but, e.g., $y_t$ denotes an element, and then again $\theta_{1:L}$ also represents a sequence. Also, P4L126 is an abuse of notation with slightly confusing wording, such as “the predictive posterior distribution over all possible designs”, whereas the predictive distribution(s) are over future \textit{outcomes}. This is in no way different than the posterior predictive in Bayesian (non-linear or linear) regression, where the posterior predictive is conditioned on the training data set and the set of (unlabeled) predictors available at test time. Hence, I struggle to understand the need for the convoluted abuse of notation, but I may be missing something. Also section 4.1 suddenly starts using bold font for vectors, which was not the case in the preceding sections.

- Figure 2 is not particularly informative for the data flow, as it does not clearly communicate weight sharing, input-output operations and dependencies (left panel); the right panel comes out of the blue and is not well explained (i.e., what are the elements on the “left” and on the “top”); the description below on P6 does indeed disambiguate the idea behind the construction of the masks, but I believe it is best when figures support and enhance the text and not vice versa.

- Overall, I feel that Section 4.1 is the weakest link in the paper, and I believe the authors can think about optimizing the ratio of details dispersed between the main text and the appendix. For instance, there is no need to reiterate established transformer-based computations, but it could be helpful to explicate the construction of the masks, the representation types (e.g., vectors, sequences of vectors,...?), and the precise partitioning of the components into keys, queries, and values.

- According to my understanding, none of the contender methods in the experiments is an amortized method. Wouldn’t some of the existing amortized BED methods (e.g., as highlighted in the Related Work) make for suitable benchmarks, despite not optimizing for future decisions?

- The topic of model misspecification is never mentioned in the paper, even though the comprehensive review paper [1] states that it remains a major unsolved issue in BED and in amortized Bayesian inference more generally [2]. I believe this should also be acknowledged in the current paper and the authors can potentially think about quantifying the impact of model misspecification in a small ablation study in the final version of the manuscript.

I am happy to discuss these points with the authors and increase my score if they are addressed / clarified.

[1] Rainforth, T., Foster, A., Ivanova, D. R., and Bickford Smith, F. (2024). Modern Bayesian
429 experimental design. Statistical Science, 39(1):100–114.

[2] Schmitt, M., Bürkner, P. C., Köthe, U., & Radev, S. T. (2024). Detecting Model Misspecification in Amortized Bayesian Inference with Neural Networks: An Extended Investigation. arXiv preprint arXiv:2406.03154.

**Questions:**

- Perhaps section 2 can be organized in a way to avoid singleton nested subsection (i.e., 2.1.1)?

- P4L130: Isn’t there also an assumption that decision are optimal only if there is no model misspecification (i.e., that we are working with the posterior of the “true” model)?

- Are there any practical disadvantages of assuming a diagonal Gaussian predictive distribution? Can complex models induce multimodal or highly correlated predictive distributions that?

**Limitations:**

The authors openly discuss the current limitations of their approach.

---

> ### Author Rebuttal · Authors · 2024-08-07
>
> Thank you for your detailed review and the valuable comments. We address your remarks and questions below.
>
> **Weaknesses**
> > 1. Some notational confusion can be avoided…
>
> Thanks, we will improve the notations according to your suggestions in the revised paper. Specifically, we will replace $h_t$ with $h_{1:t}$ to better denote a sequence of elements. And we will consistently use bold font for vectors across the whole paper.
>
> > 2. P4L126 is an abuse of notation with slightly confusing wording, such as “the predictive posterior distribution over all possible designs”.
>
> We apologize for the confusing wording and the lack of a clear explanation of the predictive distribution. Please refer to the global response for a detailed answer.
>
> > 3. Figure 2 is not particularly informative for the data flow ... the right panel comes out of the blue and is not well explained.
>
> We will improve Figure 2 to better help readers understand our work. Specifically, we will use distinct modules to more clearly visualize the data embedding block (e.g., designs in different sets share the same embedder). We will also refine the Transformer block to highlight the attention mechanisms between different sets. For the causal mask on the right panel, we will remove the elements with complex notation on the left and top, replacing them with set names for better clarity.
>
> > 4. Overall, I feel that Section 4.1 is the weakest link in the paper, and I believe the authors can think about optimizing the ratio of details dispersed between the main text and the appendix.
>
> Thank you, we will revise. To address your specific points, we will provide a more detailed explanation of the construction of the masks, along with the improvements to the masks in Figure 2 as mentioned above. As stated in our response to the first question, we will improve the notation of some elements to enhance reader comprehension. Finally, we will clarify the elements involved in the attention mechanism. Specifically, we concatenate all sets together and use self-attention to obtain a single attention matrix, then use masks to determine the dependencies between different sets.
>
> > 5. According to my understanding, none of the contender methods in the experiments is an amortized method. Wouldn’t some of the existing amortized BED methods make for suitable benchmarks?
>
> We agree that it is important to compare with an amortized method. We added PFNs4BO [1], an amortized optimization framework, as a new baseline for our top-$k$ experiments. Please refer to the global response for the experiment results.
>
> As for why we did not choose to compare with any amortized BED method before: Amortized BED mainly refers to DAD [2] and its subsequent works. These methods cannot be directly extended to our tasks for two reasons. First, DAD is a deterministic policy suitable only for continuous design spaces, whereas our tasks such as decision-aware active learning involve discrete data with covariates and their decisions. Second, even though there are subsequent works [3] extending DAD to discrete design spaces, these methods require an additional model or post-processing to make the final decision, making a direct comparison with our method inappropriate without significant alterations to DAD-based frameworks.
>
> > 6. The topic of model misspecification is never mentioned in the paper.
>
> That’s a very good point. We fully agree on the importance of model misspecification which has received significant attention recently in some areas, such as Simulation-Based Inference, but it has not been sufficiently studied in BED (beyond the handful of papers cited in [4]). Most BED works assume that the models are well-specified. In our work, model misspecification can indeed occur during the inference stage. Additionally, the utility might also shift during the deployment phase, potentially influencing the model's performance. In addition to detecting model misspecification, tackling the problem of robust experimental design under model misspecification would also be very interesting. We will add this discussion and outline it as an important area for future work.
>
> **Questions**
> > 7. Perhaps section 2 can be organized in a way to avoid singleton nested subsection?
>
> Agreed; will do.
>
> > 8. P4L130: Isn’t there also an assumption that decision are optimal only if there is no model misspecification?
>
> Yes, it is a basic common assumption in the field of BED [1, 2]. We will explicitly state this assumption in the revised manuscript.
>
> > 9. Are there any practical disadvantages of assuming a diagonal Gaussian predictive distribution? Can complex models induce multimodal or highly correlated predictive distributions?
>
> That’s a good question. Our TNDP follows the common practice in the neural processes literature [5] of using independent (diagonal) Gaussian likelihoods. If modeling correlations between points is crucial for the downstream task, we can replace the output with a joint multivariate normal distribution (similar to GNP [6]) or predict the output autoregressively (similar to AR-CNP [7]). For modelling multimodal predictive distributions, we could replace the Gaussian head with a mixture-of-Gaussians head. These modifications can be easily implemented in the TNDP architecture. We will mention this in the Discussion.
>
> **References**
>
> [1] Müller et al. (2023). Pfns4bo: In-context learning for Bayesian optimization. *ICML*.
>
> [2] Foster et al. (2021). Deep adaptive design: Amortizing sequential Bayesian experimental design. *ICML*.
>
> [3] Blau et al. (2022). Optimizing sequential experimental design with deep reinforcement learning. *ICML*.
>
> [4] Rainforth et al. (2024). Modern Bayesian experimental design. *Statistical Science*.
>
> [5] Garnelo et al. (2018). Conditional neural processes. *ICML*.
>
> [6] Markou et al. (2022). Practical conditional neural processes via tractable dependent predictions. *ICLR*.
>
> [7] Bruinsma et al. (2023). Autoregressive conditional neural processes. *ICLR*.

---

> > ### Comment · Reviewer_WHLG · 2024-08-09
> >
> > I thank the authors for their response, further evaluation, and clarifications. I will keep my positive score.

---

> > > ### Author Response · Authors · 2024-08-12
> > >
> > > Thank you for your time and consideration. We're glad to hear that you keep your positive score.

---

### Official Review · Reviewer_Zp5w · 2024-07-17

**Soundness:** 3
**Presentation:** 3
**Contribution:** 3
**Rating:** 6
**Confidence:** 4

**Summary:**

This paper tackles an important problem of designing experiments in a way that directly optimizes downstream decision-making tasks, going beyond just inferring parameters of interest. The authors make several valuable contributions:

1. They introduce the concept of Decision Utility Gain (DUG) to quantify how much an experimental design improves the expected utility of the downstream decision.

2. They propose a novel neural architecture called the Transformer Neural Decision Process (TNDP) that amortizes both the experimental design selection and the approximation of the predictive distribution needed for decision-making. This unified amortized framework is a key innovation.

3. The authors develop a non-myopic training objective that looks beyond just the immediate decision utility to account for effects of the current design on future rewards.

4. Empirically, they demonstrate TNDP's effectiveness over traditional methods on various tasks like active learning, hyperparameter optimization, showing it can find informative designs and make accurate downstream decisions.

In summary, this work makes valuable conceptual and technical contributions to the area of Bayesian experimental design by pioneering decision-aware amortized methods. It opens up new research directions for further enhancing real-world decision-making via optimized experimental data acquisition.

**Strengths:**

- The paper presents a novel problem formulation by introducing the concept of Decision Utility Gain (DUG), which shifts the focus of experimental design from reducing parameter uncertainty to directly optimizing downstream decision utility. This new perspective is a creative departure from traditional Bayesian experimental design (BED) approaches.
- The application of amortized inference techniques to decision-aware experimental design can be considered an original contribution, as it represents a new domain for these methods beyond traditional BED.
- The empirical evaluation is comprehensive, spanning diverse tasks such as active learning, hyperparameter optimization, and synthetic regression problems. The results demonstrate the consistent superiority of TNDP over traditional methods.

**Weaknesses:**

- The authors could provide a more rigorous analysis of the properties and characteristics of the TNDP architecture, such as its convergence behavior, sample complexity, and theoretical guarantees (if any) regarding the quality of the proposed designs and decisions.
- The experimental evaluation, while comprehensive, focuses primarily on synthetic and benchmark datasets. While these serve as important proof-of-concept demonstrations, the paper could benefit from including real-world case studies or applications to further validate the practical utility of the proposed framework.
- While the amortized nature of TNDP is highlighted as a key advantage, the paper could provide a more detailed analysis of the computational complexity and scalability of the proposed approach. This analysis could include factors such as the training time required for different problem sizes, the memory footprint, and the scalability of the attention mechanisms used in the Transformer architecture.

**Questions:**

- Can the authors provide a more in-depth theoretical analysis of the Decision Utility Gain (DUG) concept, including its relationship with existing concepts like Value of Information (VoI) or Information Gain (IG)?

- Have the authors explored the sensitivity of TNDP's performance to different hyperparameter choices, such as the discount factor α used in the non-myopic objective? If so, can they share insights into this analysis?

**Limitations:**

- The authors mention the use of a basic REINFORCE algorithm for training the query head, which can lead to unstable training, especially in tasks with sparse reward signals. While they suggest the use of more advanced reinforcement learning methods as a potential solution, a more detailed discussion on the specific challenges faced during training and the trade-offs involved in selecting different RL algorithms would be beneficial.
- The authors mention that their model is trained on a fixed-step length, assuming a finite horizon for the experimental design process. A discussion on the limitations of this assumption and the potential difficulties in extending their approach to infinite horizon or open-ended experimental scenarios would be valuable.

---

> ### Author Rebuttal · Authors · 2024-08-07
>
> Thank you for your positive assessment of our work and the points you raised. In the following we address your questions and points raised.
>
> **Weaknesses**
> > 1. The authors could provide a more rigorous analysis of the properties and characteristics of the TNDP architecture, such as its convergence behavior, sample complexity, and theoretical guarantees (if any) regarding the quality of the proposed designs and decisions.
>
> The primary purpose of this paper is to propose a new amortized BED framework targeted at downstream decision-making. We leverage the established REINFORCE algorithm to aid in model training. Since we did not introduce a new RL algorithm, convergence analysis can be referenced from existing works, such as the analysis provided in [1].
>
> Regarding sample complexity, for the toy example and active learning case, we generate synthetic data online. Specifically, we spent 50,000 steps to train our model and generated 16 synthetic datasets for each step. We will include these additional details in the appendix. For top-$k$ optimization, our model is trained using the publicly available HPO-B dataset, and detailed information about this dataset is already provided in the appendix.
>
> > 2. The paper could benefit from including real-world case studies or applications to further validate the practical utility of the proposed framework.
>
> We conducted a new retrosynthesis planning experiment based on real-world molecule data, please refer to the global response.
>
> > 3. The paper could provide a more detailed analysis of the computational complexity and scalability of the proposed approach. (This analysis could include factors such as the training time required for different problem sizes, the memory footprint, and the scalability of the attention mechanisms used in the Transformer architecture.)
>
> Thanks, we will add the following discussion in the revised manuscript. We have already included the rough total training time in the appendix. We will further provide descriptions of the training time and memory footprint required for each task. For example, for active learning experiments, a Tesla V100 GPU with 32GB memory was used, with an average memory consumption of 8 GB. The training time took about 10 GPU hours. Regarding scalability, our architecture is based on the Transformer, which suffers from quadratic complexity with respect to the input sequence length. This can become a bottleneck when the query set is very large. We will also include a further discussion of this issue and potential optimizations in the revised manuscript.
>
>
> **Questions**
>
> > 4. Can the authors provide a more in-depth theoretical analysis of the Decision Utility Gain (DUG) concept, including its relationship with existing concepts like Value of Information (VoI) or Information Gain (IG)?
>
> IG can be regarded as a special case of DUG. First, we define $\mathcal{P}(\theta)$ as a set of distributions that we assume contains the true posterior distribution $p(\theta|h_t)$. We then define the decision space as $\mathcal{A} = \mathcal{P}(\theta)$, and the utility function as $\log a$, where $a \in \mathcal{P}(\theta)$. The optimal action in this case will be $a^* = p(\theta|h_t)$ based on the definition of entropy. DUG can be reinterpreted as the entropy reduction when we observe a new design pair $\\{\xi, y\\}$, which corresponds to the definition of IG.
>
> > 5.  Have the authors explored the sensitivity of TNDP's performance to different hyperparameter choices, such as the discount factor α used in the non-myopic objective? If so, can they share insights into this analysis?
>
> We added an ablation study regarding the choice of the discount factor, please refer to the global response.
>
> **Limitations**
>
> > 6. A more detailed discussion on the specific challenges faced during training and the trade-offs involved in selecting different RL algorithms would be beneficial.
>
> Thanks for your suggestion. In our experiments, the REINFORCE algorithm has proven sufficient for training an effective model. However, for more complex problems in the future, REINFORCE may be prone to high variance in policy gradient estimates. If needed, RL training can be improved by using more advanced algorithms like PPO [2], but the trade-offs include introducing more hyperparameters, such as the clip ratio, and increased computational cost, which requires more tuning of the system. Importantly, our work shows that our method can be trained effectively without needing complex, ad hoc RL techniques. We will expand the discussion on these aspects in the revised manuscript.
>
> > 7. The authors mention that their model is trained on a fixed-step length… A discussion on the limitations of this assumption and the potential difficulties in extending their approach to the infinite horizon or open-ended experimental scenarios would be valuable.
>
> The finite horizon assumption is sufficient for most BED problems, as we usually operate with a limited budget for experimental designs. However, for more complex BED problems, such as long-term medical trials, extending our approach to an infinite horizon setting could be valuable. Potential challenges include increased instability during training and higher computational costs. We will expand the discussion on these aspects in the revised manuscript.
>
> **References**
>
> [1] Zhang et al. (2021). Sample efficient reinforcement learning with REINFORCE. *AAAI*.
>
> [2] Schulman et al. (2017). Proximal policy optimization algorithms. *arXiv:1707.06347*.

---

### Author Rebuttal · Authors · 2024-08-07

We thank the reviewers for their thoughtful comments and suggestions. We are glad to see that all reviewers have a positive view of the paper. Specifically, the reviewers agreed on the following strengths of the paper:
* **Relevance**: Zp5w: “tackles an important problem”. Ctfm: “relevant and interesting topic”. 7z19: “This is an innovative paper with high practical relevance”.
* **Novelty**: Reviewers Zp5w, WHLG, and Ctfm agree that “the proposed architecture is novel”.
* **Good presentation**: WHLG: “The paper is clearly written… a pleasure to read”.  Ctfm: “clarity of writing is excellent”.
* **Experiments**: Zp5w: “The empirical evaluation is comprehensive”. WHLG: “The results indicate superior and stable performance…”.

**New experiments and results**

* As suggested by reviewers WHLG and 7z19, we added an amortized method as a benchmark for top-$k$ optimization experiments. Specifically, we chose PFNs4BO [1], a transformer-based amortized model designed for hyperparameter optimization. The final results are shown in Figure R1 of the rebuttal PDF. Our method outperforms PFNs4BO across all four tasks, as PFNs4BO does not consider the downstream task (i.e., top-$k$ optimization). We will update the results in the revised paper.
* As asked by reviewers Zp5w, Ctfm, and 7z19, we ran an extra ablation study to evaluate the impact of the discount factor $\alpha$. When $\alpha=0$, our objective is purely myopic. We observed that, compared to other non-myopic settings, the policy did not learn to query designs effectively. This could be due to the sparse nature of rewards in this task; when the algorithm only considers immediate rewards, it struggles to learn the value of actions that lead to future rewards. We included the experimental results in the rebuttal PDF (Figure R2).
* As suggested by reviewer Zp5w, we included a real-world experiment on retrosynthesis planning. Specifically, our task is to assist chemists in identifying the top-$k$ synthetic routes for a novel molecule, as selecting the most practical routes from many random routes generated by the retrosynthesis software can be troublesome. We trained our TNDP on a novel meta-dataset, including 1500 molecules and their routes collected by our collaborators. In this task, experimental design refers to selecting a route for a novel molecule to query its score. The downstream task is to recommend the top $k$ routes based on the collected data. Due to limited time, we compared only TNDP and the random search. The results in Figure R3 of the rebuttal PDF show that the utility of TNDP is significantly better than that of random search. We will include more baselines and provide a detailed problem description in the final paper.

**Clarification to Section 2.2**

We thank the reviewers for raising thoughtful questions regarding the definition of $p(y_\Xi | h_t)$ and the utility function. We acknowledge that this section lacks some details, and we would like to provide further explanations here.

Our utility function is defined based on the measured outcomes ($y$) instead of the state of the world ($\theta$), as many downstream tasks directly rely on the predictions of outcomes for decision-making (see P4L124 in our paper as an example). It is a natural extension of the traditional definition of utility by marginalizing out the posterior distribution of $\theta$, a similar decision-theoretical setup can be found in [2]. As we are switching the belief about the state of the world (posterior) to the outcomes (posterior predictive) and to keep as much information as possible about the state of the world, we need to evaluate $\theta$’s effect on all points of the design space, thus, we define the utility based on $p(y_\Xi | h_t)$, which is a stochastic process that defines a joint predictive distribution of outcomes indexed by the elements of the design set $\Xi$, given the current information $h_t$. We formulate the decisions in terms of this stochastic process, which differs from traditional utility based on individual observations, such as those defined in [3, 4]. A familiar example of our framework may be a decision process that depends on the observed values of a Gaussian process simultaneously evaluated at a large number of points. For example, in top-$k$ optimization, the goal is to select $k$ hyperparameter settings from a predefined finite set that maximize the cumulative accuracy. In this task, estimating the predictive distribution of a single hyperparameter setting is not sufficient for making the optimal decision. We need to determine the optimal decision based on the predictive distributions of all candidate hyperparameter settings. We adhere to the standard definitions of decision theory, but the entities now are stochastic processes instead of individual observations.

Our architecture simultaneously amortizes two tasks. The first task is to amortize the predictive distribution needed for maximizing the utility during inference, which is similar to the goal of neural processes. When we can accurately predict $p(y_\Xi | h_t)$, we can make optimal decisions. For example, if we can accurately predict the outcomes corresponding to all hyperparameter settings, we can directly determine the optimal set of hyperparameters. The second task is to amortize the design of experiments. Our goal is to enable the neural network to propose more informative designs, thereby allowing more accurate prediction of the outcome and facilitating optimal decision-making.

We will include the above explanations in the revised paper.

**References**

[1] Müller et al. (2023). Pfns4bo: In-context learning for Bayesian optimization. *ICML*.

[2] Kuśmierczyk et al. (2019). Variational Bayesian decision-making for continuous utilities. *Neurips*.

[3] Lindley (1972). Bayesian statistics: A review. *Society for industrial and applied mathematics*.

[4] Chaloner & Verdinelli (1995). Bayesian experimental design: A review. *Statistical science*.

---

### Public Comment · ~Artur_Szałata1 · 2024-12-14
**Related work**

A closely related NeurIPS paper from last year is not cited:
Maraval, Alexandre, et al. "End-to-end meta-bayesian optimisation with transformer neural processes." Advances in Neural Information Processing Systems 36 (2024).

---

> ### Public Comment · ~Daolang_Huang1 · 2024-12-14
>
> Thank you for bringing this paper to our attention. While both works use a similar Transformer neural process-based architecture, the main distinction is that our work focuses on general downstream utility functions and decision-making tasks, while the meta-BO paper focuses specifically on BO objectives. Moreover, our work introduces a coupling between query and prediction heads in the training objective, where the prediction head helps compute rewards to guide the query head, providing a more integrated architecture for decision-making tasks. Thanks again -- we will cite this paper and discuss the connections in the Related Work section of the final version of our paper.

---

### Decision · Program_Chairs · 2024-09-25

**Decision:**

Accept (poster)

**Comment:**

The paper presents an amortized decision-aware Bayesian experimental design framework that explicitly prioritizes maximizing downstream decision utility. The paper also introduces Transformer Neural Decision Process (TNDP), an architecture capable of instantly proposing the next experimental design by inferring the downstream decision. The reviewers were generally positive about the submission and highlighted the novelty of the BED framework and the proposed architecture, and they appreciated the strength of the empirical results.